# How subtle changes in 3D structure can create large changes in transcription

Jordan Yupeng Xiao[1], Antonina Hafner[2], Alistair N Boettiger[1,2]*

[1]Program in Biophysics, Stanford University, Stanford, United States; [2]Department of Developmental Biology, Stanford University, Stanford, United States

**Abstract** Animal genomes are organized into topologically associated domains (TADs). TADs are thought to contribute to gene regulation by facilitating enhancer-promoter (E-P) contacts within a TAD and preventing these contacts across TAD borders. However, the absolute difference in contact frequency across TAD boundaries is usually less than 2-fold, even though disruptions of TAD borders can change gene expression by 10-fold. Existing models fail to explain this hypersensitive response. Here, we propose a futile cycle model of enhancer-mediated regulation that can exhibit hypersensitivity through bistability and hysteresis. Consistent with recent experiments, this regulation does not exhibit strong correlation between E-P contact and promoter activity, even though regulation occurs through contact. Through mathematical analysis and stochastic simulation, we show that this system can create an illusion of E-P biochemical specificity and explain the importance of weak TAD boundaries. It also offers a mechanism to reconcile apparently contradictory results from recent global TAD disruption with local TAD boundary deletion experiments. Together, these analyses advance our understanding of cis-regulatory contacts in controlling gene expression and suggest new experimental directions.

## Introduction

The genomes of many organisms have been shown to adopt a domain-like structure commonly referred to as topologically associated domains (TADs) (*Ibrahim and Mundlos, 2020*; *Jerković et al., 2020*; *McCord et al., 2020*; *Rowley and Corces, 2018*), defined as contiguous regions of the genome where intra-region 3D proximity is greater than inter-region (*Dixon et al., 2012*; *Nora et al., 2012*). When plotted as a heat map of contact frequency as a function of two genomic coordinates, TADs appear as boxes on the diagonal at specific genomic coordinates. They can be detected by a range of distinct techniques, including methods relying on proximity ligation like 3C/Hi-C (*Jerković et al., 2020*; *Kempfer and Pombo, 2020*; *McCord et al., 2020*; *Rowley and Corces, 2018*), ligation-free sequencing methods like GAM (*Beagrie et al., 2017*) and SPRITE (*Quinodoz et al., 2018*), microscopy methods such as ORCA (*Bintu et al., 2018*; *Mateo et al., 2019*), and even live-cell measurements like DAMC (*Redolfi et al., 2019*). Recent microscopy experiments have shown that TADs are a statistical property that emerges from a population of cells dynamically exploring multiple conformational states, rather than a static structure found in all cells (*Bintu et al., 2018*).

Despite broad agreement on the existence of TADs, their role in transcriptional regulation has recently become a subject of controversy as different recent results have been interpreted to support or refute a functional link (*Finn and Misteli, 2019a*; *Ghavi-Helm et al., 2019*; *Mir et al., 2019*). For example, many TAD boundaries demarcate regions of co-expressed genes and separate differentially expressed genes, suggesting that they play a role in cis-regulatory specificity (*Long et al., 2016*; *McCord et al., 2020*; *Spielmann et al., 2018*). Opposing this interpretation, it has been noted that the quantitative difference in interaction frequency is generally small: inter-TAD contacts often reach only half the frequency of intra-TAD contacts, raising questions of how such small

*For correspondence:
boettiger@stanford.edu

**Competing interests:** The authors declare that no competing interests exist.

differences could explain such clear separation of expression (*McCord et al., 2020*). The disruption of TAD boundaries by cis-mutation has been proposed as the causal driver of the substantial alterations in gene expression for a variety of disease conditions, a conclusion supported by animal models (*Dowen et al., 2014*; *Franke et al., 2016*; *Hnisz et al., 2016*; *Lupiáñez et al., 2015*) reviewed in *Long et al., 2016*; *McCord et al., 2020*; *Spielmann et al., 2018*. However, global disruption of TAD boundaries by depletion of boundary-associated factors led to few substantial changes in gene expression in cell culture models (*Cuartero et al., 2018*; *Nora et al., 2017*; *Rao et al., 2017*; *Schwarzer et al., 2017*; *Stik et al., 2020*; *Zuin et al., 2014*), though such depletion does obstruct the ability of macrophages to respond to inflammatory stimuli (*Cuartero et al., 2018*; *Stik et al., 2020*). Genetic experiments have made it clear that enhancers act on promoters only *in cis*, suggesting a requirement for physical proximity that would explain the functional role of TADs (*Furlong and Levine, 2018*). However, recent microscopy experiments uncovered little correlation between enhancer-promoter (E-P) proximity and nascent transcription activity at well-studied loci (*Alexander et al., 2019*; *Mateo et al., 2019*) reviewed in *Finn and Misteli, 2019b*. This surprising disagreement has led to various speculative explanations, such as action at a distance through condensates with diameters of tens of molecules (*Alexander et al., 2019*; *Benabdallah et al., 2019*; *Cavalheiro et al., 2021*; *Heist et al., 2019*; *Lim and Levine, 2021*).

We wondered whether these apparent contradictions can be reasonably reconciled with known biochemical mechanisms. To answer these questions, we examined simple biophysical models of proximity-dependent E-P communication grounded in known physical laws (the chemical master equation [CME]). Here, we identified biochemical mechanisms whose relevance for the spatial organization of the genome has not previously been considered and showed how the said mechanisms provide a simple reconciliation of all three apparent contradictions. We discuss the implications for this revised view of contact-mediated cis-regulation of gene expression for interpreting experimental results.

## Results

### A transcriptional response hypersensitive to changes in contact frequency

To better understand the connection between E-P contact frequency and transcriptional behavior, we started with a quantitative analysis of recently published super-resolution microscopy data (*Mateo et al., 2019*). These experiments used Optical Reconstruction of Chromatin Architecture to observe the 3D path of chromatin through two neighboring regulatory domains of the *Drosophila* Hox genes. The upstream domain contained the gene *Ubx* and its enhancers, and the downstream region contained the gene *abd-A* and its enhancers (*Figure 1A*). In wildtype *Drosophila* embryos, interactions between the two domains were less frequent than within the domains, producing two clearly distinct TADs in population average data (*Figure 1A*). Deletion of only a few kilobases of sequence at the border of these TADs resulted in a clear increase in interaction between the domains, including between *Ubx* enhancers and the *abd-A* promoter. However, interaction frequency increased only by a factor of 1–2.5-fold across the population (*Figure 1A*). Moreover, while the structures were easily distinguished at the population level, the structures of single cells were more variable (*Figure 1—figure supplement 1A, B*), with 25% of individual wildtype cells showing some cross-TAD border mixing between the *Ubx* enhancers and the *abd-A* promoter compared to 50% of mutant cells (*Figure 1—figure supplement 1C*). The change in gene expression, as assayed by single-molecule fluorescent in situ hybridization (smFISH), was hypersensitive to this moderate change in contact frequency, with an average fivefold increase in mRNA counts of *abd-A* (*Figure 1B*). Similar changes were also seen for *Ubx* (*Figure 1—figure supplement 2*). From these single-cell analyses, we concluded that the quantitative disconnect between weak TAD borders (see also *Figure 1—figure supplement 3*) and the large transcriptional effects of border disruption (see also *Figure 1—figure supplement 4*) is not an artifact of Hi-C or of population averaging. Instead, small differences in the frequency of contact appear sufficient to drive large changes in expression, requiring us to re-examine the textbook picture of E-P regulation (*Alberts et al., 2018*).

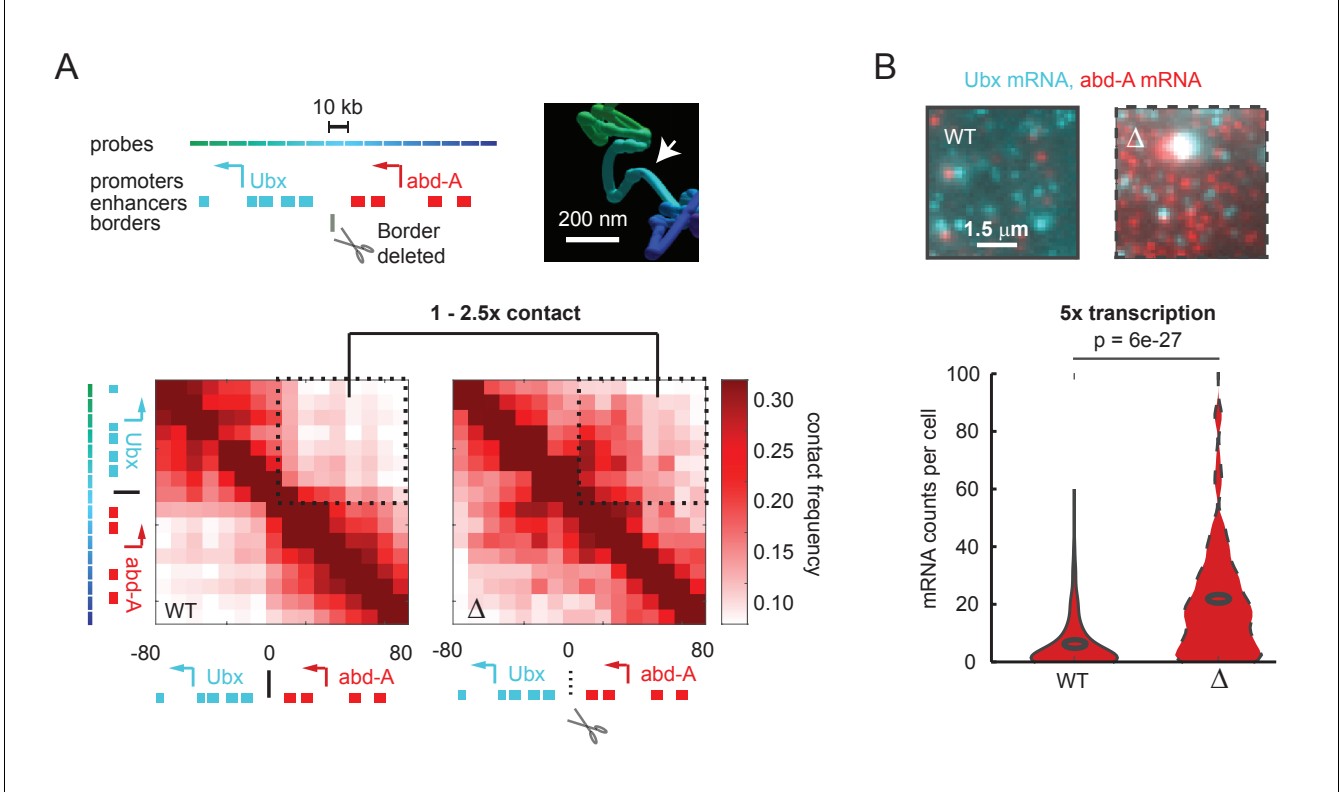

**Figure 1.** A futile cycle promoter explains super-linear transcriptional responses following subtle changes in enhancer-promoter contact. (**A**) Quantification of the minor difference in cross-border contact frequency between the *Ubx* and *abd-A* regulatory domains in wildtype and domain-border-deleted embryos, quantified by Optical Reconstruction of Chromatin Architecture. (**B**) Quantification of the major change in transcription of *abd-A* between wildtype and domain-border-deleted embryos. Raw data taken from *Mateo et al., 2019*.

The online version of this article includes the following figure supplement(s) for figure 1:

**Figure supplement 1.** Wildtype cells also exhibit cross-border contacts.

**Figure supplement 2.** Quantitative effects of topologically associated domain (TAD) border deletion on RNA expression.

**Figure supplement 3.** Calculation of topologically associated domain (TAD) insulation scores across the genome using Hi-C data.

**Figure supplement 4.** Compiled data on fold change in expression following deletion or inversion of topologically associated domain (TAD) borders.

## Modeling nonlinear transcriptional response

We set out to explore theoretical models that can explain how changes in E-P interaction frequency (3D genome folding) can lead to nonlinear effects on transcription. To do this, we used the chemical master equation (CME) approach, represented as a continuous-time Markov process (*Durrett, 2012*; *Strogatz, 2019*; *Figure 2A*), to model the discrete/stochastic nature of enhancer-promoter interactions.

Nonlinear transcriptional responses have been a subject of intense research, both experimental and theoretical (*Bergmann et al., 2007*; *Bintu et al., 2005a*; *Bintu et al., 2005b*; *Eldar et al., 2002*; *Gregor et al., 2007*; *Haskel-Ittah et al., 2012*; *He et al., 2010*; *Manu et al., 2009*; *Ozbudak et al., 2004*; *Sanchez et al., 2011*; *Ben-Tabou de-Leon and Davidson, 2009*; *Zinzen and Papatsenko, 2007*; *Zinzen et al., 2006*; *Bentovim et al., 2017*). While this previous modeling work has focused on connecting transcriptional output to changes in transcription factor (TF) concentrations and not to E-P contact, we used it as a starting point to build intuition for how to obtain a hypersensitive response. Additionally, as CMEs and continuous-time Markov processes may be less familiar to some readers, we first applied it to a familiar problem: a model of the molecular mechanisms that render transcription hypersensitive to changes in nucleoplasmic TF concentration. All models in this work start with the common framework of transcription (more details in Materials and Methods): TFs

bind an enhancer, the enhancer interacts with a promoter, and the promoter state determines transcriptional output. For simplicity, we assume that E-P interaction and binding of General Transcription Factors (GTFs) occur at a much faster rate than TF-enhancer binding, in which case transcription rate is simply proportional to the fraction of time the enhancer is bound (see Materials and Methods for a discussion on relaxing this assumption).

Consider first an enhancer with only one TF-binding site for factor $A$. The enhancer has two states: unbound and TF-bound (*Figure 2B*). It transitions from unbound to bound with a probability per unit time proportional to the number of molecules of $A$: $k_{on}A$, and transitions from bound to unbound with fixed probability per unit time $k_{off}$ (*Figure 2B*). This stochastic process can be compactly expressed by writing down the corresponding *stochastic matrix*, **M**, associated with the continuous-time Markov jump process (*Figure 2B*). Here, $P_{ij}(s,t)$ is the probability the system is in state $i$ at time $s$ given that it was in state $j$ at time $t$. The time evolution of **P** is governed by the forward Kolmogorov equation (*Figure 2A*). Standard probability theorems for Markov processes enable us to compute the complete stochastic dynamic behavior of this system, frequently through straightforward matrix operations (*Durrett, 2012*). While trivial cases like this example may be solved without this matrix formalism, the formalism will become helpful for more complex models and the matrix operations will remain identical to those introduced here. Of immediate interest, we can see a stationary state exists because **M** is irreducible and recurrent. And by solving for the normalized kernel of **M**, the stationary probability that the system is in the TF-bound state (state 2) is $\pi_2$ = $A/(k_{off}/k_{on} + A)$ (*Figure 2C*). From this result, we see, for any value $k_{on}$ or $k_{off}$, the probability that the enhancer is bound (and therefore whether it can activate transcription) exhibits only sublinear responses to $A$ (i.e., is never hypersensitive) (*Figure 2C*).

Now consider a minor variation: an enhancer that has two binding sites for factor $A$ (*Figure 2D*). For simplicity, let the two sites be identical, so the system can be described with only three states: (1) no sites bound, (2) either bound, or (3) both bound. If only the state with both sites bound is transcriptionally active, we are interested in the probability the system is in state 3, which at stationary state is $\pi_3$ = $A^2/((k_{off1}k_{off2})/(k_{on1}k_{on2}) + k_{off2}/k_{on2}A + A^2)$ (the normalized kernel of the matrix **M** in *Figure 2D*). From this equation, we observe that for the special case where $k_{off2}/k_{on2} < k_{off1}/k_{on1}$, this system is hypersensitive to $A$ (*Figure 2C*). The stronger this difference, the more the system approaches the behavior of $f(A) = A^n/(c^n+A^n)$, which is the Hill equation: a sigmoidal curve, starting at zero and reaching a maximum value of 1 as $A$ increases, with an inflection at $A = c$ (*Figure 2C*, *Figure 2—figure supplement 1*). As $n$ increases, the gentle curve sharpens into a step function, where $A = 0$ for $A < c$ and $A = 1$ for $A > c$. Thus, the Hill equation provides a convenient way of parameterizing a sigmoidal response in terms of its critical threshold $c$ and its sensitivity, $n$ (*Figure 2—figure supplement 1*).

We next wondered if a simple adaptation of this model, replacing TF interactions with enhancer-enhancer interactions, could explain the hypersensitive transcriptional response to changes in genome structure (*Figure 2E–F*). In this scenario, a small change in the initial interaction frequency increases slightly the probability that two enhancers interact, much like how a small change in TF concentration improves the probability two TFs bind the same enhancer (*Figure 2G*). Mutual affinity between these enhancers stabilizes their interaction with the promoter, much like mutual affinity between TFs stabilizes their interaction with an enhancer. In the hypersensitive regime, a minor initial change in contact frequency (x-fold), due to disruption of the TAD border, leads to a major change in E-P contact frequency (y-fold > x-fold), due to cooperativity in forming the hub, which could in turn produce a major change in transcription (*Figure 2F, G*).

Unfortunately, this model is immediately rejected by available data. Upon perturbation of TAD boundaries, we (*Figure 1A*) and others do not observe the appearance or loss of cooperative enhancer contacts of great enough magnitude to explain the change in transcription (*Figure 1B*). These and other recent multi-contact data have revealed the existence of cooperative three-way interactions (*Allahyar et al., 2018*; *Bintu et al., 2018*; *Oudelaar et al., 2019*, *Oudelaar et al., 2018*). Yet even with these cooperative effects, the fold change in measured contact frequency is not as large as the fold change in transcription before and after TAD border deletion.

## Promoter futile cycles and hypersensitive response

We thus sought to identify other mechanisms that could give rise to hypersensitivity to E-P contact frequency. Inspired by the hypersensitive cell-signaling pathways dependent on futile cycle

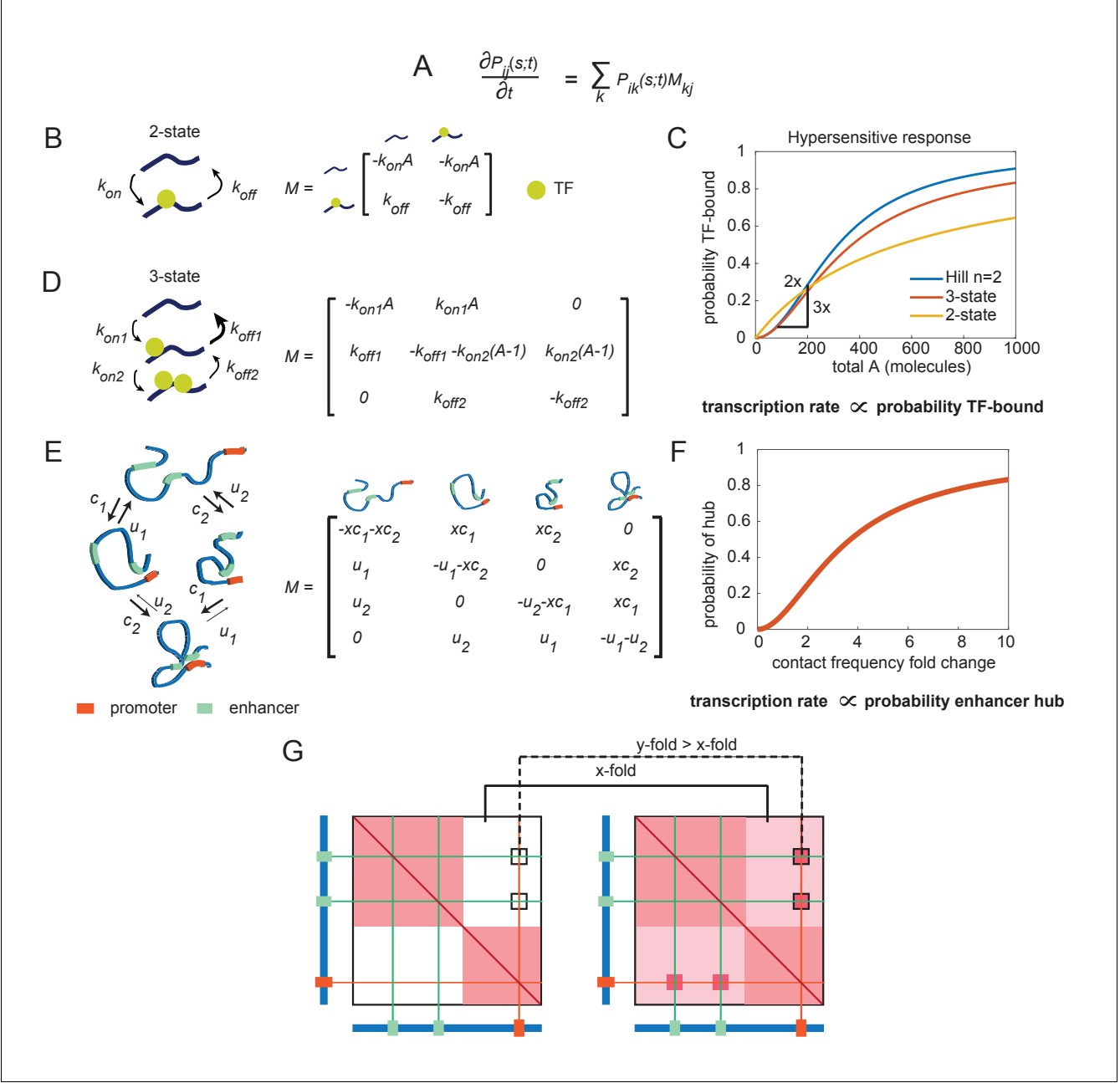

**Figure 2.** The emergence of hypersensitivity in simple stochastic transcription models. (**A**) A stochastic model for the evolution of the chemical system, which specifies how the probability of observing the system in state $i$ at time $s$ given it was in state $j$ at time $t$ evolves in time (the forward Kolmogorov equation). $M$ is a transition matrix for the continuous-time Markov system. (**B**) Cartoon illustration of chemical states for the 2-state enhancer model and the corresponding transition matrix $M$, as a function of the concentration of transcription factor (TF) activator, $A$. (**C**) The probability of TF binding as a function of molecule count, for the 2-state and 3-state systems. The Hill equation with a coefficient $n = 2$ is shown for comparison. (**D**) As in (**B**), but for the 3-state enhancer model with two potentially cooperative binding sites. (**E**) Cartoon depiction of state space for the enhancer cooperativity model and corresponding matrix $M$ for the Markov system/chemical master equation model. The DNA is denoted by a blue polymer, the positions of the enhancers by green segments, and the promoter region by an orange segment. Contacts form with probability $c_1$ and $c_2$ for enhancers 1 and 2, respectively, and separate from loop configuration with probability $u_1$ and $u_2$. (**F**) Probability of being in the two-loop state as a function of the fold change in loop frequency, for a selection of rate constants chosen to demonstrate the existence of a hypersensitive response. (**G**) Cartoon depiction of the expected change in contact frequency according to the cooperativity model presented in (**E**). Enhancer cooperativity leads to the formation of specific loop interactions.

The online version of this article includes the following figure supplement(s) for figure 2:

**Figure supplement 1.** Parameterization of the sigmoidal Hill equation.

competition between phosphatases and kinases (*Ferrell, 2012*; *Huang and Ferrell, 1996*; *Qiao et al., 2007*), we considered a mechanism in which the promoter is not waiting passively for an enhancer signal, but rather is engaged in a futile cycle of accumulation and removal of factors that favor transcription. We propose two chemically explicit realizations of this model to establish concrete examples.

The first is the 'condensate version,' in which a ubiquitously expressed GTF, such as PolII itself, is allowed to accumulate at individual promoters. Additional molecules join the condensate and dissolve back into the nucleoplasm at rates $r$ and $g$, respectively. Condensates with $n$ molecules already at the promoter provide $n$ chances per interaction for a newly arrived molecule to be captured, such that capture is more likely at larger condensates (*Figure 3A, B*). For simplicity, we assumed that the transcription rate of the gene is proportional to the size of the condensate (see Materials and methods for a discussion on relaxing this assumption). As in other futile cycle signal transduction systems, these competing processes of addition and removal happen in all cells at all times—they are not dependent on interactions with an enhancer or the presence of cell-type-specific TFs. Promoters where condensates dissolve faster than they form will generally be silent, but retain a non-zero probability of transitioning to the active state. Cell-type-specific/enhancer-dependent gene activation arises as follows: each time an active enhancer contacts the promoter, a single additional molecule is transferred to the condensate. We denoted the probability of contact per unit time as $e$, which is a property of the 3D structural dynamics of the domain. While enhancer contact is not explicitly required for promoter activation, we will show that this simple model still exhibits rapid and relatively homogenous responses to enhancer activation/deactivation. For the moment, we assumed the enhancer is already activated by the necessary TF binding and requires only looping to influence transcription, allowing us to focus on the effects of 3D structure. The relevant states in the chemical master equation are shown in cartoon form in *Figure 3B*, along with the stochastic matrix of the corresponding Markov jump process (*Figure 3C*). In the interest of simplicity, we have introduced a maximum cluster size as a fixed parameter. The effects of relaxing this assumption are discussed in Materials and methods.

This simple model version includes a regime in which condensate size (a proxy for transcription) is hypersensitive to changes in 3D structure, as illustrated by simulations of the Markov jump process (*Figure 3D*). (Specific numerical parameters for the simulations shown in *Figure 3D* and all following figures may be found in Materials and methods.) In this regime, at low E-P contact frequencies, condensates dissolve faster than they form at the promoter, and most of the population has no PolII bound. As E-P contact frequency approaches the hypersensitive regime, a higher percentage of the population binds multiple PolII molecules, but most are still unbound (*Figure 3D*). At this point, the fold change in contact is still greater than the fold change in transcription, and the system is in a transcriptional regime largely unaffected by structural change. However, a further increase in the E-P contact frequency tips the balance, such that most of the population shifts from an unbound to a mature condensate state. This is the hypersensitive regime—a small (2-fold) increase in $e$ resulted in a large (10-fold) increase in the average amount of PolII per promoter, and thus on transcription. Further increases in contact frequency have sublinear effects on transcription output.

To investigate the generality of this hypersensitive regime, we conducted a sweep of model parameters (*Figure 3E–G*). Normalizing to association probability per unit time, we characterized the response as a function of maximum cluster size, $c_{max}$, and the dissolution constant, $g$. We parameterized the response by fitting the average condensate size vs. enhancer loop rate to a Hill function and plotting the sensitivity ($n$), critical threshold ($c$), and max value of that function ($v$) (see *Figure 2—figure supplement 1*). We saw that sensitivity increased with cluster size, and hypersensitive behavior required cluster size greater than 2 (*Figure 3E*). In addition, the PolII off-rate had to be not too small, or the stability of the off-state was too low to permit hypersensitivity, and not too large, or the mature condensates were never stable (*Figure 3F*). A higher dissolution constant resulted in greater sensitivity up until the point where $g$ was so high that the cluster state was no longer reached, as can be seen in the max-value plot (*Figure 3G*). As expected, increasing the max cluster size linearly increased the cluster size obtained at the maximal E-P looping rate (*Figure 3G*). The critical threshold increases as a function of $g$ as well, also to the point where high $g$ prevented the system from ever reaching the max cluster size (*Figure 3F*). For larger $c_{max}$, larger values of $g$ can be used without preventing access to this state, allowing higher critical thresholds and higher sensitivity (*Figure 3E, F*).

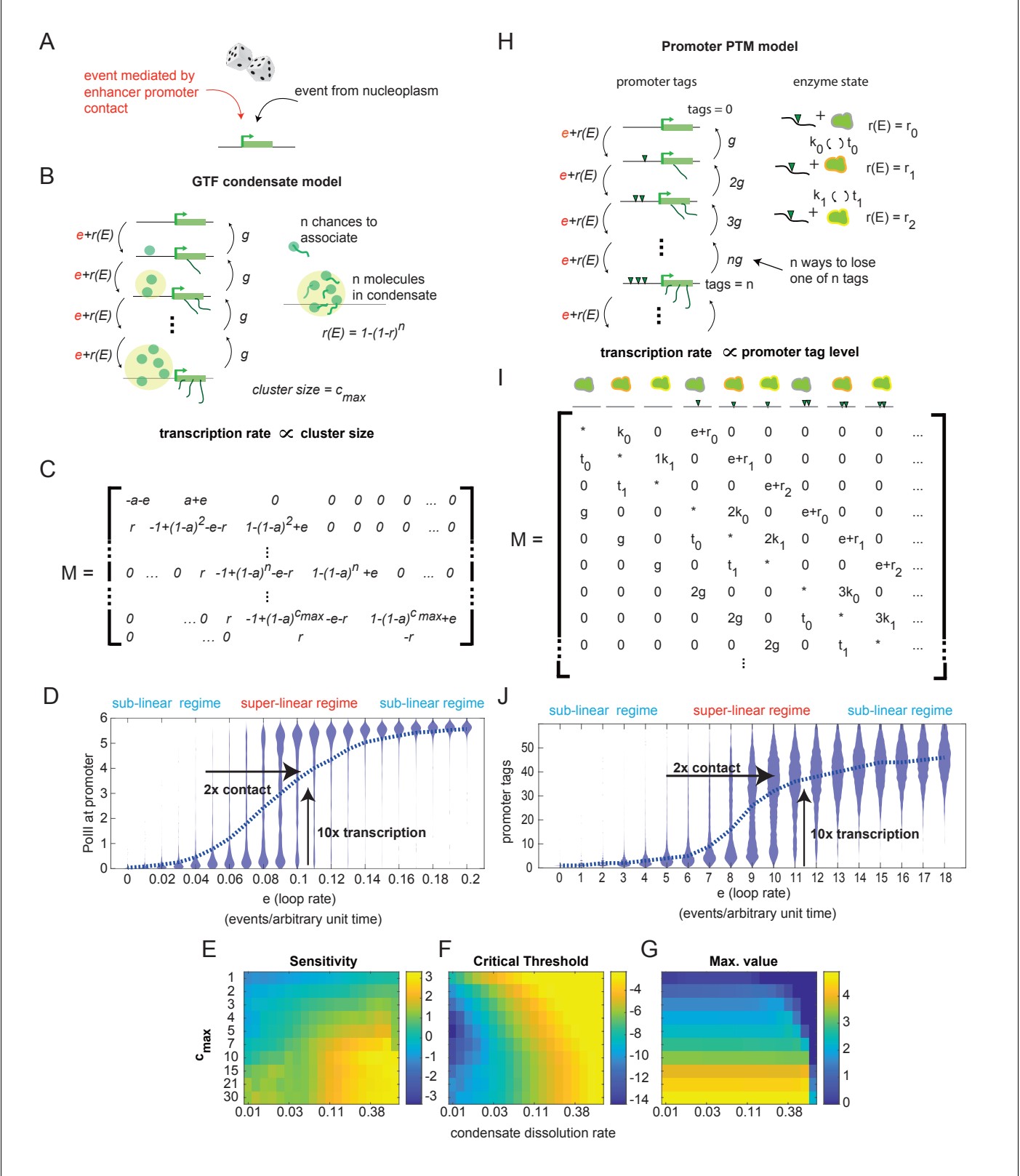

**Figure 3.** Systems with hypersensitive transcriptional responses to structural perturbation. (**A**) Generalized cartoon depiction of probabilistic events that affect promoter activity. (**B**) Schematic of the general transcription factor (GTF) condensate version of the futile cycle model. (**C**) Markov matrix corresponding to the GTF model version depicted in (**B**). (**D**) Simulated condensate size distribution as a function of enhancer-promoter looping rate $e$.

*Figure 3 continued on next page*

*Figure 3 continued*

Each violin plot represents a single simulation with fixed *e*. The superlinear regime reflects where fold change in transcription (linearly proportional to condensate size) is greater than corresponding fold change in *e*. (E–G) Sensitivity, critical threshold, and maximum value of the GTF system as a function of condensate dissociation rate and max cluster size. (H–J) Same as in (B), (C), and (D), but for the post-translational modification (PTM) model version. Numerical differences in model outputs between D and J, reflect primarily differences in parameter choices rather than intrinsic differences between the model versions, as seen also in the parameter sweeps (E-G, *Figure 3—figure supplement 1*).

The online version of this article includes the following figure supplement(s) for figure 3:

**Figure supplement 1.** Minimal conditions for hypersensitivity in the futile cycle model.

While we have described the condensate model version using PolII as an example, any variety of GTF (or even combination thereof) could play this role in the cell. Recent experimental work supports the formation of condensates of GTFs such as PolII, Mediator, and BRD4 at the promoters, (*Chong et al., 2018*; *Cho et al., 2018*; *Sabari et al., 2018*; *Lu et al., 2018*), and provides experimental evidence for enhanced capture/reduced release (*Chong et al., 2018*) as expected for phase separating condensates. The biological significance for transcription of such condensates remains disputed.

The second realization of the futile cycle model is the 'post-translational modification (PTM) version' (*Figure 3H*). It is based on accumulation of PTMs, 'tags,' deposited by constitutively expressed enzymes in the vicinity of the promoter, and does not require any condensate of molecules to form (*Figure 3H*). We assumed that the enzyme can physically interact with deposited tags, which allosterically stimulate its effective addition rate from $r_0$ to $r_1, r_2, \ldots, r_n$, where $n$ is the total number of stimulated states. For simplicity, we focused on the minimal case of $n = 2$ (i.e., the enzyme has only two stimulated states). The probabilities of enzyme-tag association and dissociation are proportional to the amount of tag existing at the promoter, with respective rate constants $k_n$ and $\tau_n$. The promoter also has an intrinsic rate of tag removal $g$. Finally, as in the condensate version, cell-type-specific regulation occurs through physical interaction with an enhancer that happens at rate $e$, determined by the 3D chromatin structure. Each E-P interaction adds one tag. The corresponding stochastic transition matrix is shown in *Figure 3I*. Note that in enumerating the states of the system we tracked all possible combinations of enzyme stimulation states and promoter tag levels.

The PTM futile cycle model version can also achieve a hypersensitive regime in which small fold changes in E-P contact lead to large fold changes in transcription (*Figure 3J*), around the experimentally observed range (*Figure 1A, B*). This hypersensitive behavior suggests that the relatively weak effects of most TAD boundaries (*Figure 1—figure supplement 3*) may still have important consequences for the control of gene expression. While the higher-dimensional parameter space of the PTM version is more cumbersome to scan comprehensively and less practical to plot, the minimal conditions for hypersensitivity can be derived by considering the deterministic limits of the stochastic system (see *Figure 3—figure supplement 1* and Materials and methods).

Thus, the requirements for hypersensitivity to structural change in the PTM version are conceptually similar to those of the condensate version. The tag-removal rate must be neither too small, or the system will activate too easily, nor too large, or the system will never be able to fully activate. This is analogous to the dissolution rate restrictions in the condensate version. The rate of accumulation of tags must be larger for systems with a larger number of tags to start with, which is achieved when $n_{max} >= 2$ and $r_0 < r_1 < r_2$ in the PTM version. In the condensate version, a conceptually similar bias is achieved through the larger capture probability of the larger condensates. As long as these conditions are met, the system admits hypersensitivity E-P loop rate $e$.

## Promoter-specific properties can create an illusion of E-P specificity

We next investigated how differences in promoter-specific rate constants in these model versions affect E-P responsiveness to TAD boundary removal. We considered two promoters that differ by a factor of 2 in $g$ (the condensate dissolution or tag removal rate), but are otherwise identical. Both start the simulation in a low-transcription state and experience the same twofold increase in E-P interaction at the start of the simulation, simulating the effect of TAD border disruption (*Figure 4A*).

For both versions of the model, Promoter 1 exhibited a large (greater than twofold) increase in transcription, yet Promoter 2 remained almost unaffected in response to the twofold perturbation of

the E-P interaction frequency (*Figure 4B*). Because of its higher intrinsic affinity for the condensate dissolution or tag removal machinery, Promoter 2 was still in the sublinear, low-transcription-response regime (*Figure 4B*). Further increases in E-P interaction would be sufficient to drive it into a high-transcribing state like Promoter 1 (due to the sigmoidal behavior explored above, *Figure 3E–G*). Quantitative differences in the simulation results largely reflect the particular parameter regime of the model (*Figure 4B*). Note that 'low-transcribing state' and 'high-transcribing state' refer to the qualitatively distinct behavioral modes of the system shown in *Figure 4*. It is this qualitative switch and the relative magnitude of the change that is of interest to us, not the numerical values. The total number of tags or molecules in the condensate between the low and high state may vary between different simulations without affecting the existence of a hypersensitive regime or the ability for a single parameter like tag removal to shift the system from the hypersensitive to the sublinear-low or sublinear-high regimes. These simulations illustrate that the apparent specificity ('lock and key') may be an illusion resulting from different tipping points among promoters, which arise from regulating accumulation/depletion of TF condensates or promoter-associated PTMs (*Figure 4C*). The promoter-specific response observed in these simulations follows from sigmoidal sensitivity to E-P contact frequency described above, which exist across a broad range of parameters.

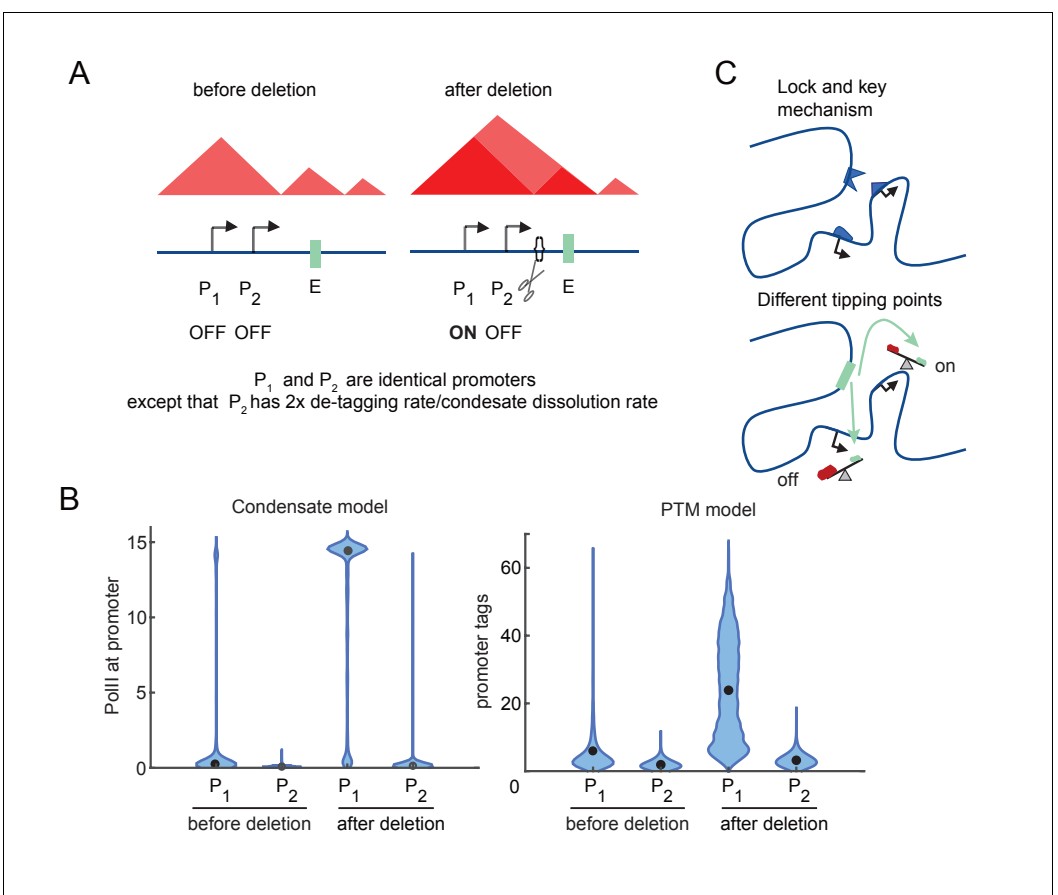

**Figure 4.** Differential sensitivity to enhancer-promoter (E-P) contact can lead to an illusion of specificity. (A) Cartoon depiction of two simulated promoters undergoing the same change in enhancer contact frequency after a topologically associated domain (TAD)-border deletion. (B) Violin plots of the changes in transcription rates (which are linearly proportional to promoter tag levels) after a twofold increase in the E-P contact rate for each promoter, simulating the border deletion depicted in (A). (C) Cartoon representation of the 'lock and key' explanation of E-P specificity, contrasted with the 'tipping point' explanation suggested by results in (B).

## Hysteresis in promoter activation leads to a unique response at different time scales

While experiments that perturb chromatin structure by deleting TAD borders have reported large (over 8-fold) changes in gene expression (*Cavalheiro et al., 2021*; *McCord et al., 2020*; *Figure 1— figure supplement 4*), experiments that globally abrogated TADs by depleting cohesin observed few genes changing expression by more than 2-fold (*Rao et al., 2014*; *Zuin et al., 2014*; *Figure 5A, B*). As these experiments differ by orders of magnitude in the time scale at which transcription changes are assessed (due to technical limitations), we wanted to explore the temporal behavior of the futile cycle promoters by assessing the effect of a TAD boundary deletion/fusion at an early time point relative to a late time point.

We found that, in addition to exhibiting hypersensitivity, both the condensate and PTM model versions readily exhibited *hysteresis*. Hysteresis means that the system has memory such that its behavior depends not only on its current state but also the preceding states (*Figure 5C, D*). For example, the transition from the high-transcribing state to the low-transcribing state occurs at different E-P contact frequency depending on whether the system started in the high or low regime initially (*Figure 5C, D*). In the continuum limit, as shown with the PTM model version, even at $t = infinity$ promoters that started in the high-tag ('ON') state transitioned to the low-tag ('OFF') state at a different value of $e$ than promoters starting in the low-tag state transition to the high-tag state (*Figure 5—figure supplement 1*), making it clear this hysteresis is an emergent property of the dynamic feedback and not a trivial delay. In the discrete stochastic case, the memory of the starting state is not permanent. As there is always a finite probability of tag removal or condensate shrinkage, there is always a finite probability that $n$ such loss events occur consecutively before a gain event, and in infinite time, such an event is guaranteed to happen, so no promoter will stay active forever. The length of the memory is determined by the difference in the average rate of gain vs. the average rate of loss in the respective high and low states. However, because of the hysteresis in the system, we observed qualitatively distinct sensitivities of the average transcription rate to small changes in E-P interaction at short time scales than we saw at long time scales (*Figure 5E*). At short time scales, the effect of a twofold change in contact frequency was minor across the whole range of E-P loop frequency compared to the change observed at long time scales in the hypersensitive regime (*Figure 5E*).

To explore this further, we examined the distribution of transcription rates in simulation for both the condensate and PTM model versions for promoters experiencing a twofold change in the E-P rate (simulating the effect of TAD loss by border deletion or cohesin depletion) (*Figure 5F, G*). We chose model parameters that positioned these promoters in the hypersensitive regime at long time scales (as in *Figures 4* and *5*; see Materials and methods). As the absolute kinetic rate constants for these model versions are not known empirically, the units of time in the simulation are arbitrary. We defined 'early' as the amount of simulation time required for a population of active promoters to transition into the non-transcribing state when the enhancer is deactivated (e.g., bound by repressors). Empirical measures of such time vary substantially among genes, but generally lie on the minutes-to-hours time scale. This scale corresponds with the time used in the cohesin-degron experiments and is distinct from the many-cell-generations/days required for testing genetic boundary deletions. We defined 'late' time points as one or more orders of magnitude longer than 'early.' With these definitions in mind, we examined the difference in simulated transcriptional response at different times. Both the condensate and the PTM simulations exhibited a weak (less than twofold) change in transcription in most of the simulated cells of the population by the early time point, even though this was sufficient for enhancer deactivation to lead most of the population to drop into the low expressing state (*Figure 5F, G*). As before, the parameter of interest is the relative change in transcription rate (tag concentration or condensate size), not the numerical value.

## Transcription and E-P contact may be largely uncorrelated in single cells

Having examined the temporal dynamics at the population scale, we turned to the dynamic behaviors in single cells. We asked whether a promoter driven by contact-dependent enhancer activity and futile cycles, as described above, would be expected to show correlation between E-P looping and transcription at the single-cell level (*Figure 6A*). From our simulated cells, we computed the odds ratios for observing transcription in a window of time given observation of contact in that

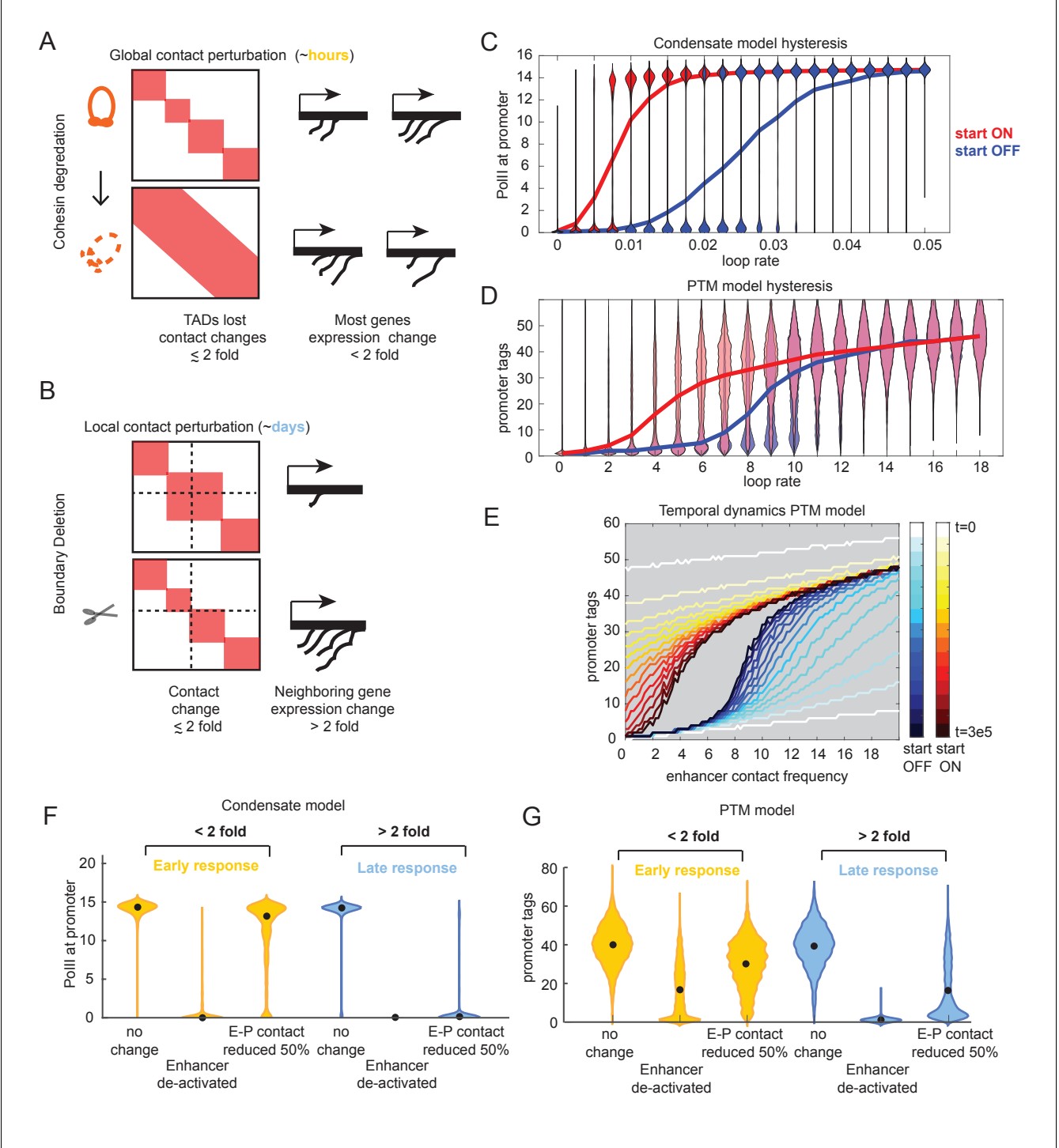

**Figure 5.** Transcriptional effects of contact perturbation depend on experiment time scale. (**A**) Schematic of global cohesin loop disruption experiments. (**B**) Schematic of individual topologically associated domain (TAD) boundary deletion experiments. (**C**) Violin plots of transcription as a function of enhancer contact, as in **Figure 4**. Blue and red violin plots represent cell populations that start in the OFF (blue, zero-molecule condensate) or ON state (red, condensate of six molecules). Blue and red lines connect the median behavior from the corresponding simulations. (**D**) Same as in (**C**) but for the post-translational modification (PTM) model version (OFF = 0 tags, ON = 30 tags). (**E**) Median transcription rate for populations starting in 'OFF' or 'ON' states, observed at different times after initialization of the simulation as indicated by the color legend. (**F**) Simulation of futile cycle promoter transcription rate as a function of time under different enhancer perturbations for the general transcription factor (GTF) condensate model version. As described in the text, 'early' is the amount of time it takes for the promoter to switch off following enhancer deactivation, and late is an order of magnitude or longer later. (**G**) Same as in (**F**), but for the PTM model version.

*Figure 5 continued on next page*

*Figure 5 continued*

The online version of this article includes the following figure supplement(s) for figure 5:

**Figure supplement 1.** Phase portrait exploration of hysteresis.

window of time, in either the condensate (*Figure 6B*) or PTM (*Figure 6C*) version of the model. Across the different regimes of E-P contact frequency, the odds ratios were statistically greater than 1, but only by a small margin of 0–30%, paralleling our recent empirical observation (*Mateo et al., 2019*; *Figure 6D*). These simulations demonstrate that contact-dependent mechanisms of E-P regulation can be still consistent with empirical data in which many transcribing cells lack contacts and *vice versa*. Since the frequency of E-P interaction only tips the scales in the existing futile cycles of promoter modification, rather than acting as a triggering event for transcription, a strong correlation is not expected. The odds ratio still remains different from 1, however, as promoters that are experiencing a high frequency of E-P contact are more likely to transcribe, increasing the probability the two events co-occur.

To further study the relation between E-P contact and transcription, we examined contact frequency in the stochastic models relative to the timing of transcription events. Such temporally ordered analysis requires live imaging data. Pioneering work by Alexander and colleagues observed no significant change in E-P distance across any time scale relative to detected transcription bursts, which was interpreted to refute a contact-dependent model of gene enhancer-mediated regulation (*Alexander et al., 2019*; *Figure 6E*). For a moderate number of simulated bursts (~500), we also found no detectable increase in proximity at any time scales relative to that of the burst (*Figure 6F, G*). This is surprising at first since by construction transcription is dependent on contact. Because the transition also depends on promoter-intrinsic addition and removal events of tags or GTFs, which also occur stochastically with variable timing, the dependence was obscured by stochastic noise. Thus, simulations involving a similar number of total cells as experiments also lacked detectable correlation between the fraction of time spent transcribing and average E-P distance or contact rate for individual cells, same as observed in the experimental data (*Figure 6—figure supplement 1A, B, D, E*). Substantially increasing the number of simulated cells (~20,000) uncovered weak but statistically significant correlations in each of these cases (*Figure 6H, I* and *Figure 6—figure supplement 1C, F*), similar to the observations from the *Drosophila* data (*Figure 6D*). Similar results were observed with either the condensate or PTM version of the futile cycle. Thus, these models show that enhancers could regulate genes like *Sox2* or the Bithorax genes in a purely contact-dependent manner, even without strong correlation between contact events and transcription at the single-cell level.

## Discussion

### Paradoxes and apparent contradictions arising from recent experimental results are reconciled by a minimal model of promoter activity

Here, we set out to understand one paradox: what molecular mechanisms, if any, could enable subtle changes in chromatin structure to lead to large changes in transcription? We developed a futile cycle model and described two possible biochemical realizations/versions that both exhibit hypersensitivity and hysteresis: TF condensates and PTM accumulation. These two distinct formulations of promoter behavior highlight the common features required for a promoter to become hypersensitive to minor changes in E-P contact frequency. Both versions propose the existence of a promoter that accumulates or loses a molecular species, the abundance of which correlates with transcriptional activity of the promoter. In both model versions, accumulation and loss proceed in futile cycles, whether or not the activating signals at the enhancer are present. For a broad range of parameter values, this futile competition renders the promoter sensitive to the activation state of its associated enhancer, and in some regimes hypersensitive to small changes in contact frequency with the active enhancer. While both TF condensates and PTM accumulation have been extensively studied empirically, their role in 3D transcriptional regulation by enhancers has received less attention in modeling, hithertofore. Using the two concrete examples, we showed that the general futile cycle model

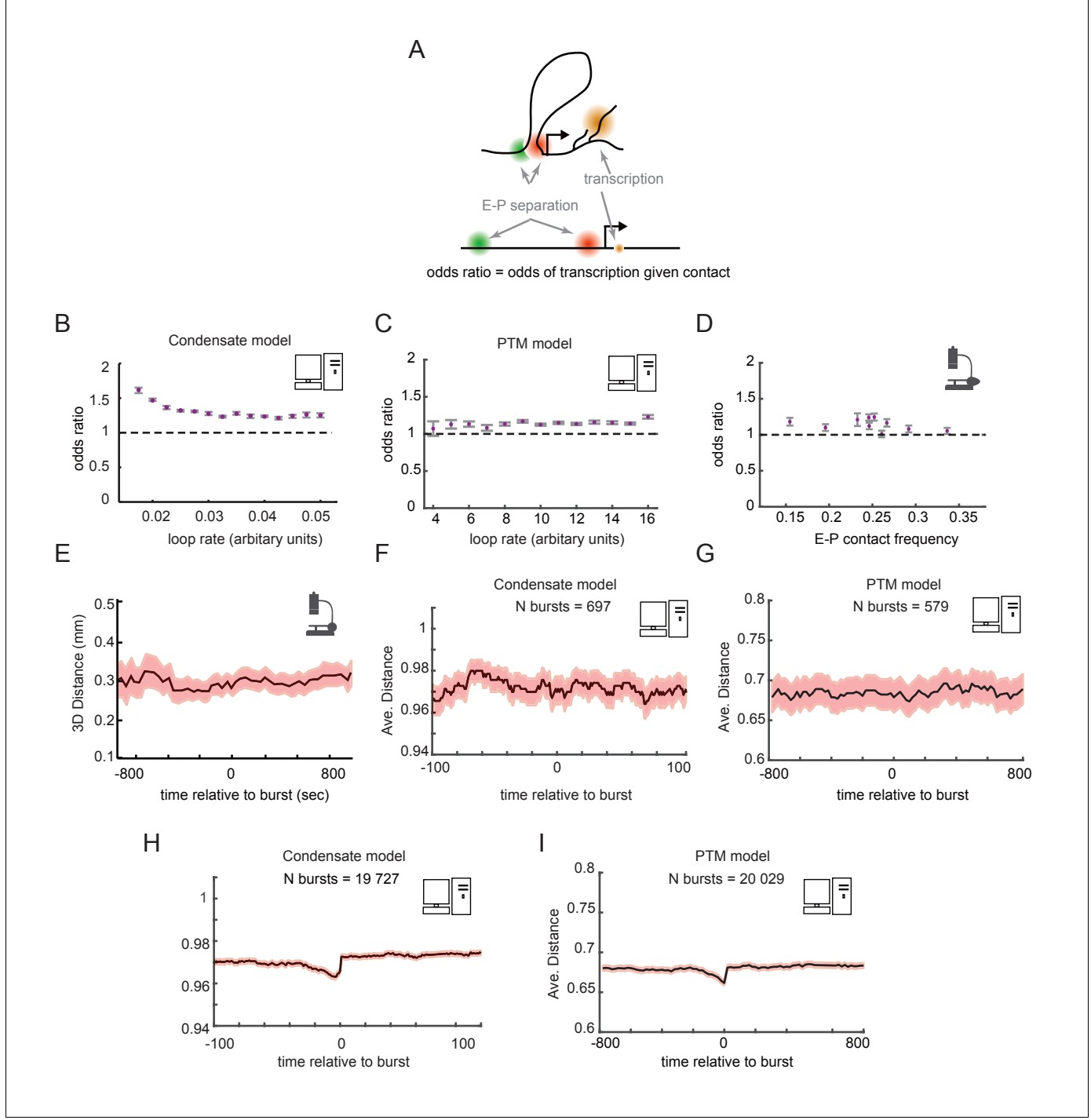

**Figure 6.** Enhancer-promoter (E-P) contact events may exhibit minimal correlation to transcription events in single cells. (**A**) Schematic of the textbook expectation of correlation between E-P contact and transcription. (**B, C**) Odds ratios for observing nascent transcription given E-P proximity, from simulations of the condensate and post-translational modification (PTM) model versions. (**D**) As in (**B**) and (**C**), but for actual data from recent super-resolution imaging experiments (*Mateo et al., 2019*). Error bars show standard error from bootstrapping. (**E**) Average measured distance between enhancer and promoter probes relative to the time of transcriptional bursts, reproduced from Figure 6E of *Alexander et al., 2019*. (**F, G**) Average E-P distance from simulations of both versions of the futile cycle model, where 0 represents contact and 1 represents no contact, relative to time transcription burst, modeled as a tag concentration > 20. Shaded region denotes ± standard error of the mean. (**H, I**) As in F and G, with sample size increased to 40-fold.

The online version of this article includes the following figure supplement(s) for figure 6:

*Figure 6 continued on next page*

*Figure 6 continued*

**Figure supplement 1.** Comparison of experimental and simulation results for the post-translational modification (PTM) version of the model on the correlation between enhancer-promoter (E-P) distance and time spent transcribing.

provides an unexpected explanation of three recent, controversial results. Here, we discuss these three controversies, identify which features of the model offer explanation of the controversy, and identify experiments which may test these features in the future.

## Some promoters may be insensitive to contact perturbations, but still activate in a contact-dependent manner

The observation that many TADs harbor genes whose spatial-temporal patterns of expression differ substantially across development poses an obstacle to the hypothesis that 3D chromatin structures like TADs contribute to regulatory specificity of gene expression (*Kvon et al., 2014*; *Schwarzer and Spitz, 2014*; *Symmons et al., 2014*). For these patterns to differ, enhancers within the TAD must be able to regulate some promoters and not others even though all experience a similar frequency of E-P interaction. A simple and common explanation of these differences is to conclude that contact frequency differences do not generally play a role in E-P specificity (since two promoters seeing similar contacts respond the same). Similarly, many enhancers are known to 'hop over' intervening promoters (*van Arensbergen et al., 2014*)—activating expression of more distal genes without notably affecting the transcription of more proximal ones, such as Shh limb enhancer (*Lettice et al., 2003*) and several FGF8 limb enhancers (*Marinić et al., 2013*) or the snail shadow enhancer in flies (*Perry et al., 2010*). In most of these cases, the more distal gene experiences less frequent contact in available measurements. Such cases have been used to argue that chemical mechanisms determine E-P specificity, and as a corollary, that structural differences that affect E-P contact frequency are of little biological relevance (*Ghavi-Helm et al., 2019*; *Kumar et al., 2020*; *Mir et al., 2019*).

Our simulations indicate that such conclusions are premature. A futile cycle promoter exhibits not just hypersensitive response to contact interactions, but also substantial regimes of sublinear or insensitive response. Promoter-specific differences in the intrinsic addition or removal of tags (or the recruitment of GTFs to a promoter condensate) can render one promoter unresponsive to an enhancer that activates a second promoter, even though the two experience similar absolute interaction frequency. Indeed, the unresponsive one may even experience a higher interaction frequency (e.g., because it is physically closer). It is thus premature to interpret such results as evidence of biochemical specificity or to dismiss a role for chromatin structure. This is notable since transgene assays indicate most regulatory elements will promiscuously activate most promoters if placed close enough (*Arnold et al., 2013*; *Furlong and Levine, 2018*; *Kvon, 2015*; *Kvon et al., 2014*). We do not argue either that one should dismiss a role for chemical specificity—several previous experiments suggest that an enhancer that strongly activates one promoter may only weakly activate a different one, and that a second enhancer may have reversed preferences (*Haberle et al., 2019*; *Hong and Cohen, 2021*). However, we do propose that existing data still supports a model in which 3D structure is of critical importance for E-P interactions genome wide. If certain perturbations show no effect, it may be because the promoter is in the insensitive rather than hypersensitive regime.

The model also suggests further experiments to test this mechanism. In particular, promoters that exhibit hypersensitive responses to certain perturbations, such as increased contact upon insulator removal, should be relatively insensitive in other regimes, which could be tested by secondary mutations. For example, increasing the contact frequency between *Ubx* enhancers and the *abd-A* promoter following deletion of the border element, by removing some of the 30–60 kb that separates them, is predicted to have sublinear effects on promoter activity since the hypersensitive regime was already crossed following the initial deletion.

A more comprehensive test of this prediction of the models would be an experiment that measured gene expression from a reporter construct, while systematically varying the separation between an enhancer and promoter across a wide range of distances, from most proximal, until too far away to have any effect. Here, E-P contact frequency would be proportional to E-P genomic distance. The models predict that there is a threshold distance, below which the promoter is active and above which it is inactive. The response around this threshold is thus hypersensitive, and further

changes at either side of the threshold are sublinear. This experiment is predicted to exhibit the sigmoidal response seen in the models: two sublinear regimes separated by a hypersensitive regime.

Such an experiment has recently been published with *Drosophila* enhancers, and results are consistent with the model prediction—decreasing E-P contact frequency by increasing E-P distance within a TAD produced a sharp drop in expression (*Yokoshi et al., 2020*). Similarly, manipulations of the *Sox9* locus, which progressively increased the contact frequency between *Sox9* enhancers and the adjacent *Kcnj2* gene by deleting first the TAD border between them and then further shortening the distance, also exhibited just such a nonlinear response (*Despang et al., 2019*). Most notably, systematic variation of the distance between the *Sox2* promoter and one of its enhancers, both inserted into an ectopic site, reveals a strikingly hypersensitive dependence of transcription on E-P contact frequency (*Zuin et al., 2021*).

## Understanding global disruption of TADs

Examination of the time sensitivity of the model provided a possible explanation of a second set of controversial experiments. It was reported in 2017 that rapid removal of cohesin from the genome substantially altered global contact patterns, removing TADs and so-called 'loops' or corner points from contact maps genome wide (*Rao et al., 2017*). Surprisingly, this structural perturbation resulted in few transcriptional changes, with hardly any gene changing nascent transcription levels by more than a factor of 2. This apparently contradicted results in earlier and concurrent reports, in which disruption of single TAD boundaries resulted in substantial change in expression to the local genes. While it is important to note that the experiments were conducted in different cellular backgrounds, our modeling work demonstrates that a simple explanation may exist. Twofold or smaller differences in contact frequency in the model show significant hysteresis and will retain their initial transcription rate for many hours after such perturbation, but this memory is lost on longer time scales. Transcription was assayed only hours after the contact changes in the degron experiment, which may explain why little change was detected. In contrast, experiments that genetically deleted border elements assayed transcription days and many cell generations later, at which point some promoters saw major changes in expression.

Notably, TAD boundaries have been globally weakened or removed through a variety of alternative methods, including degradation of CTCF, deletion of the cohesin-associated factor NIPBL, protease cleavage of cohesin, and siRNA (*Cuartero et al., 2018*; *Nora et al., 2017*; *Schwarzer et al., 2017*; *Stik et al., 2020*; *Zuin et al., 2014*). A direct comparison between the studies is complicated by the variations in cell type and genetic background used. Yet, by and large, transcriptional assays performed at longer intervals after first detection of E-P contact disruption exhibit greater transcriptional changes, consistent with the explanation suggested by these simulations.

How can we connect modeling time scales with the experimental measurements? While too little is known about the detailed chemical kinetics in vivo to confidently extrapolate these results to absolute time scales, it is clear there is a fast regime and a slow regime, with moderate change looping rates associated with a slower time scale. Available in vivo data on recruiting epigenetic modifiers to promoters predominantly shows significant response on the time scale of hours (*Braun et al., 2017*; *Hathaway et al., 2012*; *Ho et al., 1996*), though subtle responses can be detected with stronger recruitment in as little as 15 min in some studies (*Braun et al., 2017*). If the promoter tags of our PTM model version are realized by such epigenetic mechanisms (discussed below), our results are consistent with the expectation that a decreased or increased contact frequency over only 6 hr will be insufficient to change the promoter state, while a substantially longer period such as the multiple days/generations between deletion and measurement can give the promoter enough time to change states. Less is known about condensate kinetics, whose very existence and reliable detection in vivo remain controversial. Nonetheless, studies reporting condensates of GTFs such as Med1, PolII, and Brd4 indicate a stability that generally exceeds the imaging time (*Chong et al., 2018*; *Cho et al., 2018*; *Sabari et al., 2018*).

Our model suggests experiments to test the hypothesis that promoters are hysteretic. Deletion of essential enhancers, such as the *Sox2* control region, results in substantial decreases in gene expression, even though alterations to the interaction frequency do not. If expression were to be measured within 6 hr of enhancer deletion or inactivation, rather than many cell generations later, the model predicts the change would be much less dramatic. Though not a trivial experiment to execute, this could be attempted using CRISPR to induce the deletion in a population (or CRISPRi to

silence the enhancer), followed by fixation within a few hours and combined single-molecule RNA FISH and DNA FISH. RNA FISH would quantify RNA expression per cell, and the DNA FISH would sort out which cells actually contained the deletion.

## Transcription may be dependent on E-P contact even when the two are not correlated in single cells

It has been widely expected that active promoters should colocalize with their active enhancers, based largely on bulk population assays such 3C and H3K27ac Hi-ChIP (*Bartman et al., 2016*; *Deng et al., 2012*; *Kagey et al., 2010*; *Mumbach et al., 2017*, *Mumbach et al., 2016*; *Rao et al., 2014*; *Williamson et al., 2016*); for review see *Furlong and Levine, 2018*. Surprisingly, recent single-cell measurements found that cells with close E-P proximity were only slightly more likely to exhibit nascent transcription than those lacking contact (*Finn and Misteli, 2019b*; *Mateo et al., 2019*). It is possible the correlation between looping and expression is missed in these fixed cell assays because the loops are transient relative to the rate of transcription, such that loops dissociate by the time the nascent RNA transcript is detectable. However, recent live-cell imaging experiments suggest otherwise: live-cell tagging of chromatin proximal to the enhancer and promoter of *Sox2* coupled with live-cell measurements of nascent RNA transcription uncovered no significant correlation between E-P proximity and transcription activity (*Alexander et al., 2019*). These data have supported speculation that enhancers must be able to influence promoter activity from a distance, for example, through bridging by large protein condensates (*Alexander et al., 2019*; *Benabdallah et al., 2019*; *Heist et al., 2019*).

The futile cycle model demonstrates that it is possible for transcription to be regulated by E-P contact and yet show little temporal correlation, even in live, single-cell time-lapse data. This is because individual E-P contact events have little impact on promoter state, contributing to only marginally more promoter tags or larger condensates. However, repeated interaction events, coupled with promoter-intrinsic feedback to accumulate tags, eventually lead to activation. This explanation can be directly tested with larger data sets—with sufficient observations, a weak correlation should be detectable (as seen in the fixed cell data), since cells experiencing a higher frequency of contact are more likely to have a high level of tag and thus more likely to transcribe, even though it is not a one-to-one dependency.

A simple test of this could be performed by collecting deeper single-cell live imaging data sets—a feasible though exceedingly laborious task given the available throughput of live imaging. An order of magnitude more measurements may allow detection of the expected weak temporal correlation. A yet more direct experiment would be to take single-cell measurements of the protein state of the promoter, with simultaneous measurement of transcriptional state—for example, monitoring the level of H3K4me3 or PolII. Of course, this experiment is exceedingly difficult with current technology, which lacks reliable detection of protein state at individual promoter loci, and is further complicated by the substantial number of candidates to test for promoter accumulation signals. While dynamic measurements would lead to stronger conclusions, even in fixed cell populations, correlating promoter state in terms of protein occupancy and abundance to the RNA expression state would be informative. While out of reach of current technologies, given the rapid improvements in experimental methods for enhanced sensitivity and versatility, it is imaginable that such direct measures will one day be possible.

## Repression also supports hypersensitivity and hysteresis

For simplicity, the futile cycle model presented above assumed that the condensates or the tags that accumulated at the promoter are activating, such that transcription is proportional to tag level. However, it is equally plausible that futile cycles of repressive condensates or tags at the promoter account for hypersensitivity of some promoters. In this case, transcription can be modeled as inversely proportional to the concentration of the promoter tag; everything else about the analysis and its conclusions remains unchanged. The distal regulatory element that adds more of the tag should be termed a silencer rather than an enhancer. In response to increased contact with the silencer, the promoter would transition through a weakly sensitive ON regime, a hypersensitive bistable regime, to a weakly sensitive OFF regime. Alternatively, a model in which the enhancer removes the repressive tag from the promoter would support all three regimes.

## Conclusion

We have described and analyzed a simple futile cycle model of enhancer-contact-dependent promoter activity and presented two concrete realizations for analysis. In this view, the promoter is not merely an intermediary in the regulation of transcription. It does not only relay the activity state of the enhancer into transcriptional activity of its target gene, nor function only as an amplifier of enhancer activity to set the levels of expression. Instead, through accumulation of local tags or GTFs in a condensate, the promoter is able to integrate its history of interaction with one or more regulatory elements and respond in striking nonlinear ways to changes in these signals. In addition to its core sequence, the accumulation of other factors at the promoter, be they histone modifications or transcription factors, provides the necessary mechanism for the dynamical system to exhibit memory.

This view of the promoter as an integrator of signaling by accumulation of molecules is not new and does not posit the existence of any unknown molecular mechanisms. However, we found that a minimal model that captures the key features of such promoters can exhibit a much more complex dependency on chromatin structure than previously acknowledged. Our modeling results offer a simple explanation to controversies of major importance to the chromatin structure and transcription community. These include (1) why TAD boundaries can be weak and why insulators only mildly affect contact frequency and yet be major regulators of transcription; (2) why many genes thought to be responsive to distal enhancers are insensitive to some structural changes; (3) why global disruption of structure and acute disruption of structure have been reported to have distinct effects on transcription; and (4) why physical distance between known distal enhancers and their cognate gene does not show obvious correlation with transcriptional state in single cells.

# Materials and methods

## Transcriptional framework and terminology

The terms *enhancer* and *promoter* have fluid definitions in the literature, with overlapping empirical criteria. For clarity, we will use the following simplified definitions. *Regulated genes* are genes activated by cell-type-specific combinations of *transcription factors* (TFs), which bind regulatory DNA sequence elements called *enhancers*. The enhancer *loops* to the promoter at some frequency determined by the genome structure. The *promoter* is a different regulatory DNA element that contains the transcription start site (TSS) and binding sites for *general transcription factors* (GTFs). As they bind the promoter, GTFs can influence transcription immediately, whereas TFs, which bind enhancers, only affect transcription when an E-P loop occurs. There is no loss of generality with these definitions—a TF-binding site that is close to the TSS, rather than being described as part of the 'promoter,' is, in this view, part of an enhancer with a high interaction frequency (one which is likely never rate-limiting). Many informative prior models of transcriptional regulation (*Ben-Tabou de-Leon and Davidson, 2009*; *Gregor et al., 2007*; *Manu et al., 2009*; *Sanchez et al., 2011*; *Zinzen et al., 2006*; *Zinzen and Papatsenko, 2007*) assume that the rate of transcription is simply proportional to TF occupancy. These models are consistent with the general definition we just described, with the added assumptions that E-P contact and GTF-promoter binding are fast relative to TF binding, and that transcription itself is proportional to GTF binding. This broader definition will help us see how genome structure would be expected to impact the behavior of these models.

## Nonlinear mathematical frameworks

Two distinct mathematical frameworks have been used to model nonlinear transcriptional systems. *Mass action kinetics* models chemical entities using continuous variables representing concentrations. Interaction rates are quantified by on/off rate constants. The mathematical tools of ordinary differential equations (ODEs) can be used to study the behavior of systems under these assumptions (*Strogatz, 2019*). Alternatively, the CME models the chemical species as discrete molecules, which interact stochastically. The mathematical tools of probability theory, particularly continuous-time Markov chains, allow analysis of the behavior of such discrete systems (*Durrett, 2012*). Each chemical state of the system is enumerated, and transitions between states occur with a finite probability that depends only on the current state. The CME's assumptions are less restrictive than the ODE approach, and it allows the study of variability in stochastic systems. In the limit where the number

of discrete molecules is large, the master equation is accurately described by the corresponding continuous ODEs. Stochastic models of transcription have been instructive in understanding promoter bursting and how different chemical signaling pathways buffer or amplify stochastic variations, which in turn improves robustness or increases heterogeneity (Ycart and Peccoud 1995, Raj 2006, *Sanchez et al., 2011*, Boettiger 2013).

## Solutions to the chemical master equation/Markov jump process

Consider a system with *n* states. Note *n* is not required to be finite. Let $\mathbf{P}(t)$ be a matrix where element $p_{i,j}(t)$ indicates the probability the system is in state *i* at time *t* given that it started in state *j* at time 0. The evolution of $\mathbf{P}(t)$ is given by the forward Kolmogorov equation, $d/dt\ \mathbf{P}(t) = \mathbf{P}(t)\mathbf{M}$, where $\mathbf{M}$ is the transition probability matrix, shown in each of the model figures (see *Figures 2–4*).

The stationary distribution of $\mathbf{P}\pi$ is given by the normalized kernel of $\mathbf{M}$. In the two- and three-state models shown in *Figure 2*, we were interested only in the probability the system was in the fully occupied state, and thus inspect only the last element of $\pi$. In other models, like the condensate model, transcription was assumed to be proportional to the size of the condensate, and we must sum all the elements of $\pi$ scaled by their corresponding transcription rate for the corresponding condensate size in order to determine the expected distribution of transcription.

It is also instructive to look at solutions not at steady state. This solution is straightforward for small matrices (see *Durrett, 2012*), but for large matrices is more efficiently explored by stochastic simulation of the underlying jump process as discussed below.

## Stochastic simulation of the Markov jump process

Stochastic simulations were written in MATLAB 2020a. Executable source code to run these simulations is available here: https://github.com/BoettigerLab/model-transcription-looping (copy archived at swh:1:rev:8af1e7fcf1fa126908465558c4ef6d127000a9c8), *Boettiger, 2021*. A detailed table of parameters values and parameter descriptions is provided at the start of each script. The associated figure is indicated in the filename and in the header of the corresponding script for each simulation.

Why use stochastic simulations when analytic methods are available? While the matrix formalism can be used to solve analytically the time-dependent and stationary probability distributions for all the states of interest in both the PTM mechanism and condensate mechanism flavors of the model, we find the complexity of the expressions in full variable form does not easily facilitate interpretation. The analytic solutions in variable form require many pages just to write down, and as such do not facilitate the mental evaluation of limit behavior as demonstrated with simpler models. By contrast, the parametric approximation of the numerical results we find more intuitive.

In brief, these simulations model the chemical master equation in explicit time and discrete time steps, with discrete transition probabilities for the addition and removal of promoter tags. For simplicity, we equate promoter modification and transcription, reducing the total parameter space without loss of generality.

The number of time steps used in the simulation was explored systematically to benchmark their effects on model behavior (e.g., see *Figure 5E*). As can be seen from these simulations and inferred from the structure of the model, the system asymptotically approaches the stationary distribution, such that the difference in the distribution of states observed after $10^4$ or after $10^5$ units of simulation time is small. The simulations in *Figure 5* explored also the more transient behaviors at shorter time scales, which were benchmarked against the time it took the simulation to respond to enhancer deactivation. The initial conditions were selected to have all promoters start in the OFF state with no tags or condensate molecules, unless indicated as a 'start on' simulation. In the latter case, all promoters were initialized with a condensate or tag size equal to that observed on average at stationarity. The specific number of time steps used in each simulation and the specific initialization conditions are recorded along with the corresponding rate constants used in the supplemental code provided.

## Model assumptions and justification

### The free concentration of high abundance molecular species, like nucleotides, soluble GTFs, or tags, is unchanged by their accumulation at the promoter

RNA production consumes ATP to produce multiple mRNA molecules from a DNA template and so these models do not explicitly conserve energy or mass. This does not violate any laws of physics; other processes in the cell, not explicitly described in the model, produce (and consume) species such as ATP in such a way that mass energy in the universe is still conserved. When these components are in sufficient excess and produced at a sufficient rate, they are reasonably approximated as constants.

This assumption can be removed with no observable change to the results, save added complexity in their computation. This follows as the relative effect of a minor depletion of the bulk population, from $N$ to $N$-1, has an effect size of $1/N$ on the addition rate.

### Transcription is proportional to condensate size/tag amount

To keep the parameter list minimal and emphasize the chemical features chiefly responsible for the hypersensitivity to E-P contact frequency, we assumed transcription is proportional to condensate size and tag amount. This assumption can be replaced with a variety of more complex, more stochastic models of transcription. As long as transcription is correlated with the accumulated amount of these factors, the hypersensitivity will remain. The degree of hypersensitivity will be lessened (or require averaging over many more promoters in order to emerge) if the correlation is weak.

### Enhancers are already bound by TFs

It is straightforward to add a more complex model of the enhancer, one, for example, which cooperatively binds multiple TFs and thus whose occupancy depends on the concentration of the respective TF activators and repressors as in Zinzen 2006 and related works. One must decide which TF occupancy configurations are capable of influencing transcription via PTM or condensate recruitment. The addition then is a function not just of the loop frequency $e$, but rescaled by the probability the enhancer is in an active state(s), which is a function of the TF concentrations.

### Condensates have a fixed maximum size while promoter PTMs do not

To keep the parameter space small in the condensate model version and facilitate a systematic scan over parameters, we modeled the system with a fixed maximum condensate size. A more realistic model would postulate that this value is a stochastic number, determined by the concentration at which the GTF forming the condensate passes below the solubility critical point and stops condensing. If the concentration is always below this threshold, no condensation will be observed. For systems with concentration above the critical point, condensates form, and compete to associate out the extra molecules. Should distinct liquid condensates meet in 3D space, they will fuse, though the frequency of such events will be dramatically reduced if the condensates are tethered on chromatin as postulated in our model, as the dense packaging of these long polymer fibers limits their diffusive mobility. In the tethered regime, condensate size is limited by the competition with other condensates for the limited pool of GTF molecules and the limited lifetime of the processes permissive to formation at a given site (such as the activity of the enhancer). Importantly, the soluble pool of GTFs remains large even once condensates have formed, and thus individual GTFs will still leave and fuse with existing condensates. The tethered condensate model is analogous to beads of dew condensing on a glass window as the air cools, forcing a fraction of water molecules out of the gas phase and into the liquid phase—the surface tension keeps the beads generally in place, so they do not all condense into a single droplet. The size of the beads is limited by the amount of water available above the critical dew point (which depends on the size of the temperature drop and how saturated the air was to begin with). Individual water molecules in the dew drops continually escape back into the gas phase, and individual molecules in the air continually join the dew drops, and are preferentially captured by the larger drops. Local variations in temperature of the substrate, much like local variations in the enhancer activity of different promoters in the model, will cause variations in the growth and dissolution rates of the different droplets.

No fixed saturation for the total amount of tag is assumed in the PTM model version. Since tag removal is proportional to current abundance, for all non-zero removal rates the amount of tag will always have a finite average. Since an individual promoter can always add one more tag though, the transition matrix is infinite. As the substrates for common PTMs (phosphates, methyl groups, acetyl groups) are in high abundance and under rapid turnover in the cell, the addition of a tag to the promoter is not expected to deplete the available pool and so impact the next modification.

## Analytic inference of the parameter space of the PTM model

Standard dynamical systems theory (*Strogatz, 2019*) reveals the continuum limit version of the PTM mechanism has at most three real, critical points, one unstable and two stable, at high and low tag levels, respectively, arising from the intersection of a sigmoidal tag production curve, $p(a)$ with a linear decay $d(a)$ (*Figure 3—figure supplement 1*). Hypersensitivity in this model arises for parameter regimes in which a minor increase or decrease in $e$ leads to the loss of either the upper or lower stable point in a saddle node bifurcation, causing the system to tip into the remaining stable point. The conditions for the existence and annihilation of these stable points can all be seen from a phase-portrait analysis of the shape of the curves $p(a)$ and $d(a)$. The effects of the individual rate constants on the shape of $p(a)$ and $d(a)$ can be seen from the analytical forms of these equations (*Figure 3—figure supplement 1A, B*). The amplitude of $p(a)$ sets the maximal transcriptional difference between enhancer-induced and -uninduced states. The sensitivity and stability of the response to enhancer promoter contact is determined by the sensitivity of $p(a)$. When $p(a)$ is a gentle sigmoid, it is possible for a very small change in $e$ to transition the system from a mono-stable off-state, through the bistable regime, into the monostable on-state (*Figure 3—figure supplement 1C*). In the corresponding stochastic regime, the difference in $p(a)$ and $d(a)$ for values of $a$ left of the upper stable point determines the ultimate stability of the upper stable point. In this stochastic regime, when the difference between $p(a)$ and $d(a)$ is small, there is a non-zero probability of losing more tags than gained in a certain time window, even when $p(a)$ is on average greater than $d(a)$. Thus, hypersensitivity to $e$ is reduced/lost as the Hill coefficient of $p(a)$ approaches 1 or approaches infinity: moderate Hill coefficients in $p(a)$ produce the most hypersensitivity to $e$ (*Figure 3—figure supplement 1C–E*). Similarly, if $d(a)$ becomes too large or too small, it also limits the sensitivity, as can be seen schematically by its effect on the position of stable points (*Figure 3—figure supplement 1F*).

## Derivation of the continuum limit equations

The ODE model (*Figure 3—figure supplement 1A*) was derived from the discrete chemical master equation using mass-action kinetics as described below. The stimulated enzyme example presented below is only one implementation of the PTM version, but used here because it is the simplest and most intuitive system that recapitulates the desired quantitative behavior.

### Model parameter definitions

Consider a promoter-bound enzyme that attaches molecular tags to a promoter at some basal rate $r_0$. This enzyme can also associate with $n$ promoter-bound tags up to some number $n_{max}$, which changes the tagging rate to $r_n$. The enzyme-tag association rates $k_n$ are dependent on the current association state (e.g., cooperativity), while dissociation is assumed to be constant (for now, but can be generalized later). 'Free' tags not associated with the enzyme but still promoter-bound are removed from the promoter with a rate constant $g$.

### Reaction and rate equations

For illustrative purposes, we will start with the $n_{max} = 2$ case (later, we will show this is sufficient to capture nonlinear behavior of interest). Then, the enzyme has three 'association states' (n = 0, 1, or 2):

Stimulation of enzyme activity by tag association:

$E + a \rightleftharpoons Ea$ with rate $k_1$ forward, $\tau_1$ backward
$Ea + a \rightleftharpoons Ea_2$ with rate $k_2$ forward, $\tau_2$ backward

where $E$ is enzyme and $a$ is tag. In general, there may be up to $n$ different stimulation states:

$Ea_n + a \rightleftharpoons Ea_{n+1}$ with rate $k_n$ rightward, $\tau_n$ leftward

Tagging/detagging:

$E \rightarrow E + a$ with rate $r_0$
$Ea \rightarrow Ea + a$ with rate $r_1$
$Ea_2 \rightarrow Ea_2 + a$ with rate $r_2$
$a \rightarrow 0$ with rate $g$

In general:

$Ea_n \rightarrow Ea_n + a$ with rate $r_n$

Note: The enzyme adds a tag to the vicinity of the promoter by reacting with abundant ambient substrate or coenzyme (e.g., Acetyl-CoA). As described above, the abundance of this substrate/coenzyme is assumed to be sufficiently large that its addition to the promoter does not appreciably change the total abundance, enabling us to reduce the number of necessary parameters by accounting for this constant concentration in the production rates $r_n$. In other words, $a$ appears to be created 'from nothing,' but the requisite material pool actually exists on both sides of each reaction equation and is simply omitted for readability.

Now we can write out the rate equations:

$$\frac{d[E]}{dt} = -k_1[E][a] + \tau[Ea] \tag{1}$$

$$\frac{d[Ea]}{dt} = k_1[E][a] - \tau[Ea] - k_2[Ea][a] + \tau[Ea_2] \tag{2}$$

$$\frac{d[Ea_2]}{dt} = k_2[Ea][a] - \tau[Ea_2] \tag{3}$$

$$\frac{d[a]}{dt} = r_0[E] + r_1[Ea] + r_2[Ea_2] - g[a] + e \tag{4}$$

where $e$ is the tagging contribution from E-P contact. Assuming the total enzyme number is constant,

$$[E_{tot}] = [E] + [Ea] + [Ea_2] \tag{5}$$

Our goal is to find $\frac{d[a]}{dt}$ as a function of $[a]$ and constants, and to study the steady-state behavior of the system, that is, finding the solutions for $[a]$ when Equations (1)–(4) are all set to zero. Solving (1) and (3), then substituting $[Ea]$ and $[Ea_2]$ into (4) and (5):

$$\frac{d[a]}{dt} = r_0[E] + r_1\frac{k_1}{\tau}[E][a] + r_2\frac{k_1k_2}{\tau^2}[E][a]^2 - g[a] + e \tag{6}$$

$$[E_{tot}] = [E] + \frac{k_1}{\tau}[E][a] + \frac{k_1k_2}{\tau^2}[E][a]^2$$

$$\Rightarrow [E] = \frac{[E_{tot}]}{1 + \frac{k_1}{\tau}[a] + \frac{k_1k_2}{\tau^2}[a]^2} \tag{7}$$

Combining (6) and (7):

$$\frac{d[a]}{dt} = [E_{tot}]\left(\frac{r_0 + r_1\frac{k_1}{\tau}[a] + r_2\frac{k_1k_2}{\tau^2}[a]^2}{1 + \frac{k_1}{\tau}[a] + \frac{k_1k_2}{\tau^2}[a]^2}\right) - g[a] + e$$

$$= [E_{tot}]\left(\frac{r_0\frac{\tau^2}{k_1k_2} + r_1\frac{\tau}{k_2}[a] + r_2[a]^2}{\frac{\tau^2}{k_1k_2} + \frac{\tau}{k_2}[a] + [a]^2}\right) - g[a] + \Xi \tag{8}$$

The terms with $r_n$ represent enzyme tagging, and the term with $g$ is detagging. This nonlinear system with $n = 2$ can be solved explicitly for the steady-state values of $a$, though the solutions are less compact than the final equation, and are most easily interpreted in the phase portraits of Equation (6). Analytic steady-state solutions were calculated in MATLAB using the symbolic algebra package. Scripts to reproduce these calculations are available on our GitHub site at: https://github.com/BoettigerLab/model-transcription-looping.

### Insulation score calculations

Several variations of this metric exist, but we define it as follows: first, the contact frequency map is normalized relative to linear genomic distance by dividing each off-diagonal row by the sum of the row. This improves the interpretability of the measure as the inter-domain triangle contains more interactions further from the diagonal than the intra-domain ones (*Figure 1—figure supplement 3*). We then scan the whole genome, computing the average, normalized interaction frequency within each of the three triangular regions shown in *Figure 1—figure supplement 3*: upstream (intra-domain U), downstream, intra-domain D, and between inter-window (domain I) of query point. We report the ratio of intra-domain to inter-domain interactions for this sliding window as the insulation score for that genomic position, as indicated in *Figure 1—figure supplement 3*. This metric can be computed for a range of sliding window sizes. As long as the window size is smaller than the typical TAD and large enough not to be sensitive to measurement-error fluctuations associated with over-binning the data, the distribution of insulation scores does not depend strongly on bin size. A comparison of several different computational algorithms for identifying TADs showed greatest agreement across replicates and robustness to data normalization using this metric (*Gong et al., 2018*; *Lazaris et al., 2017*).

Balanced, Hi-C matrices from *Rao et al., 2014* were downloaded and exported using Juicebox (*Durand et al., 2016*) for manipulation in MATLAB for computing insulation scores.

## Acknowledgements

We thank Luca Giorgetti and Gregory Roth for informative discussions and members of the Boettiger lab for a critical reading of the manuscript. We thank Daniel Ibrahim for insightful and inspiring discussions about the behavior of structural perturbations near the *Sox9* locus. This work was supported by a CASI award from the Burroughs Wellcome Foundation a NIH New Innovator Award (DGM132935A) to ANB and support from the 4DN consortia (U01 DK127419). JX was supported by the Molecular Biophysics Training Program at Stanford (GM008294). AH was supported by a Walter V. and Idun Berry Postdoctoral Fellowship.

## Additional information

### Funding

| Funder | Grant reference number | Author |
|--------|------------------------|--------|
| National Institutes of Health | U01 DK127419 | Alistair N Boettiger |
| NIH | DGM132935A | Alistair N Boettiger |
| Stanford University | GM008294 | Jordan Yupeng Xiao |
| Stanford University | The Walter V. and Idun Berry Postdoctoral Fellowship Program | Antonina Hafner |
| Burroughs Wellcome Fund | CASI | Alistair N Boettiger |

The funders had no role in study design, data collection and interpretation, or the decision to submit the work for publication.

### Author contributions

Jordan Yupeng Xiao, Conceptualization, Formal analysis, Methodology, Writing - original draft; Antonina Hafner, Supervision, Validation, Writing - review and editing; Alistair N Boettiger,

Conceptualization, Software, Formal analysis, Funding acquisition, Methodology, Writing - original draft, Project administration

### Author ORCIDs
Jordan Yupeng Xiao https://orcid.org/0000-0001-8072-9341
Antonina Hafner https://orcid.org/0000-0003-4927-5227
Alistair N Boettiger https://orcid.org/0000-0002-3554-5196

### Decision letter and Author response
Decision letter https://doi.org/10.7554/eLife.64320.sa1
Author response https://doi.org/10.7554/eLife.64320.sa2

## Additional files

### Supplementary files
• Transparent reporting form

### Data availability
This is a theoretical and computational paper using published experimental data.

The following previously published datasets were used:

| Author(s) | Year | Dataset title | Dataset URL | Database and Identifier |
|---|---|---|---|---|
| Rao SSP, Huntley MH, Durand NC, Stamenova EK, Bochkov ID, Robinson JT, Sanborn AL, Machol I, Omer AD, Lander ES, Aiden EL | 2014 | A three-dimensional map of the human genome at kilobase resolution reveals prinicples of chromatin looping | https://www.ncbi.nlm.nih.gov/geo/query/acc.cgi?acc=GSE63525 | NCBI Gene Expression Omnibus, GSE63525 |

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
