## [Decision Letter]

**Acceptance summary:**

The work describes a simple theoretical model for enhancer action that explains several major controversies in the field of long-range gene regulation and the role of topologically associating domains and insulating boundaries in modulating enhancer-promoter interactions. Further, the model makes predictions that be experimentally tested. This is valuable for the field of gene regulation.

**Decision letter after peer review:**

Thank you for submitting your article "How subtle changes in 3D structure can create large changes in transcription" for consideration by *eLife*. Your article has been reviewed by 3 peer reviewers, including Job Dekker as the Reviewing Editor and Reviewer #3, and the evaluation has been overseen by Naama Barkai as the Senior Editor. The following individual involved in review of your submission has agreed to reveal their identity: Katie S. Pollard (Reviewer #2).

The reviewers have discussed the reviews with one another and the Reviewing Editor has drafted this decision to help you prepare a revised submission.

Summary:

The work describes a simple theoretical model for enhancer action that explains several major controversies in the field of long-range gene regulation and the role of topologically associating domains and insulating boundaries in modulating enhancer-promoter interactions. Further, the model makes predictions that can be experimentally tested. This is valuable for the field of gene regulation.

Essential revisions:

All three reviewers agreed the work addresses a major question in the field but also raised important issues that should be addressed. These primarily revolve around how the model is described, and the extent to which the full parameter space is explored. Further, the authors have not explored alternative models which would help clarify how surprising and robust their results are. There is also an opportunity to emphasize that the proposed model is not necessarily absolutely correct, but one of many plausible models that can produce a non-linear relationship between genome structure (enhancer-promoter contact) and transcription. Any thoughts on other models that could generate similar dynamics would be a useful discussion point.

1. The presentation of the model is unclear. It is currently present in the text, lines 110-122, in pure qualitative description. Authors define only rates in the text; definitions of other model parameters are not present. For example, E and a are not specifically defined in the text or Methods section. Since both terms "enzyme" and "enhancer" are being used and in fact "enzyme tagging" and "enhancer tagging" occur simultaneously in the model, it is not possible to say for sure when do authors call which one in the model and thus the methods section can be interpreted in different ways. Moreover, the cartoon is missing a legend confirming, which molecular player is which. The figure caption mentions only green triangles being the tags, but no other parts of the cartoon are being explained. Taken together, this makes it very difficult to verify the mechanics of the model.

– Given its centrality to the manuscript, we recommend describing the overall strategy in more detail in Results. For example, at line 124 (Pg. 4) the authors could talk about how the simulations are done, including where the variability comes from (e.g., random starting conditions vs. probabilistic events vs. different parameters).

– – The authors should provide a detailed technical description of their model directly in the text, including description of their parameters, list their constitutive equations and identify all parameters in their cartoon Figure 1C.

– Axes labels in all figures should be expressed in the parameters/variables of the model (as in Figure 6C-D) directly connecting to inputs/outputs of the model.

2. In the Methods section, it appears that in lines 577-580 of the model description, the mass is not conserved. This issue is critical to address, because when mass is not conserved the model can not be physical.

3. Xiao et al. make several key assumptions to dramatically simplify their model. Namely, it is assumed that promoter modification and transcription are equivalent and that enhancer-promoter contact influences transcription instead of transcription influencing structure. Steady-state equilibrium must also be assumed. It would be helpful if the authors explicitly stated these assumptions and provided references to support their being reasonable.

4. For clarity, it would be helpful to discuss model parameters in greater detail. First, we suggest noting which parameters shift the location of the curve and which increase the steepness of the curve. Second, we recommend including a phase diagram exploring when sigmoidal behavior and any other key model predictions arise across parameter space. In what circumstances does hypersensitivity or time lag emerge? The authors demonstrate that a narrow set of parameters is sufficient to produce a super-linear relationship between enhancer-promoter contact and transcription in Figure 6. One potential dilemma is this model's ability to explain many experimental observations by indicating that minimal changes all occur in the sub-linear regime while observable changes occur in the super-linear regime. Given that one needs specific parameters to replicate an example of the hyper-linear regime (including at least three degrees of stimulation and increasing stimulation of the successive states), it could be valuable to demonstrate how large the plausible parameter space is. Without an exhaustive search across the space of minimal parameters, it is not clear when this property emerges or how common it is within the full parameter space. The authors could vary model parameters and plot a grid visualizing behavior (e.g., steepness of the curve or Hill coefficient).

5. The authors observe hysteresis in median transcription rate as a function of enhancer contact frequency. However, the presented violin plots suggest a presence of two states, one with low and one with high transcription rates. In the intermediate regime of enhancer contact frequency, where authors report hysteresis, the violin plots show bimodal distributions suggesting coexistence of these two states. This would suggest that the system exists in and switches between two distinct states with a discontinuous transition, instead of a continuous hysteretic behavior as suggested by the median behavior.

6. There is also an opportunity to emphasize that the proposed model is not necessarily absolutely correct, but one of many plausible models that can produce a non-linear relationship between genome structure (enhancer-promoter contact) and transcription. Any thoughts on other models that could generate similar dynamics would be a useful discussion point. There are parallels to both sigmoidal dose-response curves, where drug concentration is plotted against response, and transcription factor binding curves, where free ligand concentration is plotted against the fraction bound. We recommend providing background context on these types of models or the Hill equation to illustrate why non-linear behavior is or is not surprising given the proposed model.

7. It is not totally clear why the authors decide to call their proposed approach the futile cycle model. There are similarities to other well-known models in biochemistry and biophysics that should be noted. It might make sense to simply call this a mechanistic model of cooperative promoter activation. If the authors stick with "futile cycle", the relationship between promoter activation through tags and metabolic signaling should be described in more detail.

8. At the beginning of the Discussion section authors state they will propose future experiments in each section. However, in some of the sections it is not clear what specifically authors are proposing. These suggestions should be made clearer.

9. If I understand the model correctly, the non-linearity arises because of the increase rate of tag addition when tag is already present. The authors then speculate histone modifications can be one such tag. However, there is only so many sites of modification at a promoter. Can the authors analyze how the possible range of tag densities affects performance of the model? Is the range required biologically plausible?

10. Can the authors do more analysis to explore how rapid changes in gene expression may occur (e.g. upon signaling a gene may go up within minutes)? How much more frequent does the E-P interaction need to be for rapid switch to the active promoter state? Can the authors do an analysis where they change the rates of the futile cycle upon some signal: at what time scale does transcription then change (keeping E-P frequency the same)?

11. Due to the lack of description, in many sections it is not clear what are the specific inputs and outputs of the model (e.g. Figure 2).

12. The Methods section describes the chemical kinetics of the suggested reactions and the insulation score calculations. But it is not clear how do these inform each other, how are contact-frequency maps chosen/computed and cross-referenced with the local E-P kinetics?

13. In 587-588, the index of k is 2(n+1), which equals to 2n+2, but then in the next line the following assumption is made 2n+1 → n+1.

14. The authors make assumptions that their kinetic considerations hold for n>2. What is the evidence?

15. The language of the paper is often not technically precise with qualifiers missing, which could lead to ambiguities and misinterpretations. Here are some examples:

p. 1, line 10, "difference in contact across TAD borders is usually less than twofold";

p. 1, line 17, "results from recent cohesion disruption";

p. 2, line 71, "A simple model of hypersensitivity to changes in contact frequency".

16. On p. 13, line 483, authors define Ostwald ripening as given by weak multivalent interactions; however, Ostwald ripening is a thermodynamic process. In addition, they propose that liquid condensates become larger due to Ostwald ripening, but there are also other processes that may occur, such as coalescence of condensates, which would also lead to larger condensates.

17. An alternative explanation for TAD-specific enhancer action is that an E-P interaction within a TAD (between two convergent CTCF sites), one that is brought about by extruding cohesin, is not equivalent to an interaction that occurs between two loci on either side of a CTCF site and that can be a random collision that is not mediated by extruding cohesin. In other words, two interactions can be of the same frequency but can be of a very different molecular nature. I agree that this model would not explain the results of the experiment where cohesin is acutely removed.

18. In the beginning of the introduction the authors introduce TADS. I recommend that the authors present this in a more nuanced way: compartment domains also appear as boxes along the diagonal, an issue that has led some in the chromosome folding field to be confused. This reviewer believes TADS are those domains that strictly depend on cohesin mediated loop extrusion, whereas compartment domains are not. If the authors agree, perhaps they can rewrite this section?

---

## [Author Response]

Essential revisions:All three reviewers agreed the work addresses a major question in the field but also raised important issues that should be addressed. These primarily revolve around how the model is described, and the extent to which the full parameter space is explored. Further, the authors have not explored alternative models which would help clarify how surprising and robust their results are. There is also an opportunity to emphasize that the proposed model is not necessarily absolutely correct, but one of many plausible models that can produce a non-linear relationship between genome structure (enhancer-promoter contact) and transcription. Any thoughts on other models that could generate similar dynamics would be a useful discussion point.

We thank the reviewers for these very helpful comments. We provide line-by-line responses below.

We have substantially rewritten the manuscript to address these concerns. We added a new section, “Modeling nonlinear transcriptional response”, (pages 6-11), that now provides a detailed background on the modeling approach used, including a discussion situating this work relative to prior work in the field. This section includes 1 new figure (Figure 2) and 1 new supplemental Figure (Figure 2 supplement 1). The section on the futile cycle model, “Promoter Futile Cycles and Hypersensitive Response”, (pages 11-15) provides a detailed description of the futile cycle model, also with detailed new figures (Figure 3 and Figure 3 supplement 1). To help the reader appreciate which features of the model are general while still providing a picture of what specific molecules and biochemical mechanisms might execute these general behaviours, we now present two distinct versions of the Futile Cycle model. The “PTM” version is based on the accumulation of Post-Translational Modifications at the promoter. This is the model which appeared in the first submission, though it is now presented in much more detail. The second “Condensate” version is based on the accumulation of GTF condensates at the promoter. The stochastic transition matrices for Kolomogorov’s Forward Equation are written explicitly in the figure to remove any ambiguity about the mathematical approach used, remove uncertainty about which transition probabilities link which states, and clarify what the states are in the model. This equation and some basic concepts of Markov chains and their application in the stochastic Chemical Master Equation are all introduced in the revised background. To facilitate interpretation for the general reader, we introduced cartoon schematics in addition to these stochastic matrices, which summarize the same states and transition probabilities in pictorial form instead of equations.

By describing two biochemically distinct versions (condensate formation/dissolution and PTM accumulation/turnover) and highlighting their common features which give rise to hypersensitivity, hysteresis, and ability to reproduce recent seemingly-contradictory findings, we hope to address the question of the Futile Cycle model’s generality. We hope you find this revised presentation to strike a balance between abstract generality and molecular reality (e.g. in place of abstract promoter states 1..n, we consider condensates of TFs and PTMs).

1. The presentation of the model is unclear. It is currently present in the text, lines 110-122, in pure qualitative description. Authors define only rates in the text; definitions of other model parameters are not present. For example, E and a are not specifically defined in the text or Methods section. Since both terms "enzyme" and "enhancer" are being used and in fact "enzyme tagging" and "enhancer tagging" occur simultaneously in the model, it is not possible to say for sure when do authors call which one in the model and thus the methods section can be interpreted in different ways. Moreover, the cartoon is missing a legend confirming, which molecular player is which. The figure caption mentions only green triangles being the tags, but no other parts of the cartoon are being explained. Taken together, this makes it very difficult to verify the mechanics of the model.

We agree the presentation of the model was unclear and has been completely rewritten, see pages 11-15 for the new text, and new Figure 3.

We now describe essential qualitative features of the model, and propose two different chemical mechanisms which exhibit those features. One chemical mechanism is based on the accumulation of post-translational modifications, which we renamed “PTM model version” from “Futile cycle” model in the first submission. The other model version requires no enzymes but assumes instead that TFs can form condensates at promoters (the “Condensate model version”). The Chemical Master Equation (CME) for each model is now shown explicitly in the new Figure 3. For simplicity we solve the CME by numerical simulation.

To aid in general readers unfamiliar with stochastic systems, Markov processes, or the Chemical Master Equation, we have provided a brief introduction to these systems as well (see Figure 2). We use this stochastic approach to briefly re-derive classic results on transcriptional hypersensitivity to cellular signals, and contrast that case to the focus of our work, hypersensitivity to changes in 3D structure. See section “Modeling nonlinear transcriptional response” on pages 5-11.

In addition to these substantial additions to the main text as suggested, we have substantially expanded the methods section to provide further technical details about the analytic methods, simulations, and parameter values. We also highlight two reference texts that provide the curious reader an excellent introduction to mathematical frameworks used in this study, continuous time Markov processes (Rick Durrett: Essentials of Stochastic Processes (Durrett 2012)) and Nonlinear Dynamical Systems (Strogatz 2018). The simulation parameters and model code are provided on our Github page, and we have fixed the broken link in the text (which was previously broken—a sorry oversight on our part). This includes new scripts with symbolic mathematical analyses of several of the Markov processes described in the text, along with simulations of the new condensate version and updated simulations of the PTM version of the model.

We hope this revised presentation is much clearer about the chemical processes postulated, and the mathematical approaches used in the analysis.

– Given its centrality to the manuscript, we recommend describing the overall strategy in more detail in Results. For example, at line 124 (Pg. 4) the authors could talk about how the simulations are done, including where the variability comes from (e.g., random starting conditions vs. probabilistic events vs. different parameters).

We thank the reviewers for this suggestion and have rewritten the first half of the manuscript to introduce the overall strategy in detail. Please see pages 5-15. It will now be clear that the stochasticity arises from our use of a Chemical Master Equation approach—modeling the molecular interactions as a continuous-time Markov process. As described in our introduction to the CME for non-specialists (see pages 7-8), stochasticity arises from a probabilistic treatment of molecular interactions. To aid in building an intuitive understanding of the model’s general behavior, we also make use of the continuum limit, in which the CME is reasonably approximated by a system of ODEs, allowing us to use tools from the analysis of deterministic dynamical systems. While the primary conclusions of the paper rest on the stochastic simulations, we found these limiting cases to be useful in building intuition, and understanding and predicting the model behavior. In the hopes that it will be of interest to the mathematically inclined reader, we include the detailed analytical work in the supplement. The revised text clearly flags where continuum theory is being discussed and where dynamic CMEs are being used.

We have also substantially expanded the methods, adding an explicit section on the assumptions, including the selection of starting conditions and their impact on model behavior.

– The authors should provide a detailed technical description of their model directly in the text, including description of their parameters, list their constitutive equations and identify all parameters in their cartoon Figure 1C.

This has been added (see above), see also equations added in main figures.

– Axes labels in all figures should be expressed in the parameters/variables of the model (as in Figure 6C-D) directly connecting to inputs/outputs of the model.

This has been done, and the relation between the model parameters (tag count or condensate size) have been explicitly added to the figures as well.

2. In the Methods section, it appears that in lines 577-580 of the model description, the mass is not conserved. This issue is critical to address, because when mass is not conserved the model cannot be physical.

See Equations 4 and 5. This ODE version of the model is used only for building intuition, as should now be clear.

The revised presentation of the model should now make it clear where simplifying assumptions arise (see next reply). Regarding the reservoir assumption (when some molecular species are in such excess relative to their levels at the promoter that gaining a molecule may be safely assumed to have a negligible change on the available pool), we assumed that the substrate material for the tag (acetyl groups, methyl groups and/or phosphate groups) is in considerable excess in the cell, and under continual production and destruction by a myriad of other cellular processes, such that the addition of one acetyl group to one promoter does not significantly deplete the pool in such a way as to make a measurable impact on the probability that a second acetyl be added.

3. Xiao et al. make several key assumptions to dramatically simplify their model. Namely, it is assumed that promoter modification and transcription are equivalent and that enhancer-promoter contact influences transcription instead of transcription influencing structure. Steady-state equilibrium must also be assumed. It would be helpful if the authors explicitly stated these assumptions and provided references to support their being reasonable.

The necessary assumptions have been more clearly stated when introduced in the main text.

Additionally, we have added a section to the Methods to recap all the assumptions in one place.

In particular we specify:

a) Tags are in excess such that addition of one tag does not affect the probability of the addition of the next in the PTM model. As acetyl/methyl/phosphate groups are in rapid turnover in 100s of cellular processes that occur on the timescales of transcriptional regulation and are available in uM concentrations in cells, this assumption appears reasonable. See

[https://bionumbers.hms.harvard.edu/bionumber.aspx?id=101259&ver=3&trm=acetyl+CoA&org=].

b) In the ‘condensate’ version of the model we assume that TFs such as PolII and Mediator, which make up the condensate model, are in sufficient excess such that a single molecule joining a promoter condensate (of up to a max of <30 total molecules) does not affect the probability of binding of the next. As the typical cells contain thousands to tens of thousands of such molecules, this assumption seems reasonable.

c) We assume transcription rates are linearly proportional to condensate size or tag number with a proportionality of unity. This assumption simplifies the model in such a way it is easier to understand, and can be trivially relaxed. As we are only interested in relative transcription changes, any extension of the model to assume a deterministic number of transcripts per tag per unit time immediately cancels out in the relative transcription change (and would trivially rescale the y-axis in our graphs without changing the qualitative behavior). A more realistic model may assume some stochastic probability of transcription initiation (or burst of initiation) as a function of tag number / condensate size. If the stochasticity in this process is very large, such that transcription is all-but-random and uncorrelated to tag or condensate size, then regimes of the model which predicted hypersensitive transcriptional responses to structure change will no longer show that the condensate size is still hypersensitive, and the transcriptional response is all-but random (by construction). But as long as the two are correlated (or anti-correlated) the key qualitative behavior will persist. (This result is mathematically trivial—a sigmoid with random noise added is still sigmoidal, even if averaging is required to make the form visible).

d) The revised presentation should make it clear that steady state is not assumed. We in fact explore the qualitative behavior of the model across different time scales, including the very instructive stationary states of the Markov process and the stable states of the corresponding ODEs in the continuum limit. Relevant time scales become an essential feature of this analysis, which we have elaborated on in the manuscript to describe how the time points are selected (see page 19).

e) Initial conditions are now specified in the Methods as follows (see also the source code):

Unless otherwise indicated, the promoters were all started in the OFF state, defined as 0 tags (PTM version) or 0 proteins in a condensate (condensate version). In several cases, clearly indicated in the text, we explored the memory effects at moderate time scales by starting the system in the high/ON state. In this case we used an initial condition in which all promoters started at the average tag-level at the maximum enhancer-promoter frequency examined (typically 40 tags) or the maximum condensate size (typically 6 to 20). These details have been added to the Methods, in addition to residing in the simulation code itself (for which the links should now be fixed!).

4. For clarity, it would be helpful to discuss model parameters in greater detail. First, we suggest noting which parameters shift the location of the curve and which increase the steepness of the curve. Second, we recommend including a phase diagram exploring when sigmoidal behavior and any other key model predictions arise across parameter space. In what circumstances does hypersensitivity or time lag emerge? The authors demonstrate that a narrow set of parameters is sufficient to produce a super-linear relationship between enhancer-promoter contact and transcription in Figure 6. One potential dilemma is this model's ability to explain many experimental observations by indicating that minimal changes all occur in the sub-linear regime while observable changes occur in the super-linear regime. Given that one needs specific parameters to replicate an example of the hyper-linear regime (including at least three degrees of stimulation and increasing stimulation of the successive states), it could be valuable to demonstrate how large the plausible parameter space is. Without an exhaustive search across the space of minimal parameters, it is not clear when this property emerges or how common it is within the full parameter space. The authors could vary model parameters and plot a grid visualizing behavior (e.g., steepness of the curve or Hill coefficient).

We have added the parameter sweeps as requested (see Figure 3E-G, and new Figure 3 supplement 1 and 7). We have characterized these with respect to the parameters of the Hill function (see Figure 2 supplement 1).

Our explorations identify a set of minimal conditions (parameters) that support hypersensitivity.

These are now described individually in the text:

a) Something must accumulate at the promoter.

b) Accumulation must be opposed by a removal process.

c) Accumulation must have positive feedback in such a way that promoters with more of the accumulating entity are more likely to get more than those that have little or none, up to some maximum value.

These high-level properties may be realized by a variety of molecular processes. Depending on the explicit model, they may be affected by a combination of kinetic parameters or by single parameters (see Figure 3, Figure 3 supplement 1, and Figure 5 supplement 1). We find these 3 conditions to be much more intuitive and accessible to the reader than writing k_1_ < tau_1_ and k_2_ > tau_2_ and k_2_ > k_1_, especially given that the model can be recast as a GTF condensate in which these three general principles still hold, even though realized through a different set of chemical states with distinct state-transition probabilities that are unrelated to enzyme kinetics (see Figure 3).

5. The authors observe hysteresis in median transcription rate as a function of enhancer contact frequency. However, the presented violin plots suggest a presence of two states, one with low and one with high transcription rates. In the intermediate regime of enhancer contact frequency, where authors report hysteresis, the violin plots show bimodal distributions suggesting coexistence of these two states. This would suggest that the system exists in and switches between two distinct states with a discontinuous transition, instead of a continuous hysteretic behavior as suggested by the median behavior.

The reviewer’s understanding of the model is largely correct—the system *is*bistable and the violin plots reveal the bistability. This bistability produces hysteresis, as can be seen not only in the medians (Figure 5C-E) but also in the newly added plot overlaying the violins (Figure 3D and J, Figure 5C-D). In addition to showing the violin plots, we have overlaid graphs showing the averages and/or median transcription rate across the experiment in many of our figures, as these population level quantities are the ones measured in many bulk experiments.

As should be clearer in our revised text, and from the responses above, the model is a discrete stochastic system (a CME or equivalently, a continuous-time Markov process). It proceeds in discrete jumps (e.g. from 0 tags to 1 tag, even though the mean tag count can be 0.1, 0.3 individual promoters only have integer tag values). There are no discontinuous jumps: the model, by construction, only gains one tag at a time (or one protein at a time) and similarly only loses one protein at a time. It is an emergent property of the model that a promoter that has lost beyond a critical number of tags is rather more likely to lose the rest (one-by-one) than to gain any back, and a promoter that has over the critical number is more likely to gain another. This dynamic behavior is most easily understood by considering the continuum limit of the model, which allows one to use the stability analysis for ODE models common in deterministic systems (see Strogatz: Dynamical Systems, Chapter 1-2).

6. There is also an opportunity to emphasize that the proposed model is not necessarily absolutely correct, but one of many plausible models that can produce a non-linear relationship between genome structure (enhancer-promoter contact) and transcription. Any thoughts on other models that could generate similar dynamics would be a useful discussion point. There are parallels to both sigmoidal dose-response curves, where drug concentration is plotted against response, and transcription factor binding curves, where free ligand concentration is plotted against the fraction bound. We recommend providing background context on these types of models or the Hill equation to illustrate why non-linear behavior is or is not surprising given the proposed model.

We have now added a substantial discussion of prior work on hypersensitive transcriptional responses, including the classic model of transcription-factor cooperativity which is one mechanism to create a hypersensitive response to chemical signals (like morphogens or drug dose-response), pages 5-11. This helpful addition provides an opportunity to highlight both the parallels and differences between these models and those which we present subsequently, improving the clarity of the presentation. Please see the new Figure 2 and the associated text on pages 8-11.

As the reviewer points out, the ‘Futile Cycle model’ which we renamed ‘PTM model version’ is not the only plausible model. We clarified this in text and provide another example with the ‘Condensate model version’ of how hypersensitivity to E-P contact can be achieved.

See also response 1 and 4.

7. It is not totally clear why the authors decide to call their proposed approach the futile cycle model. There are similarities to other well-known models in biochemistry and biophysics that should be noted. It might make sense to simply call this a mechanistic model of cooperative promoter activation. If the authors stick with "futile cycle", the relationship between promoter activation through tags and metabolic signaling should be described in more detail.

The new section, “Promoter Futile Cycles and Hypersensitive Response”, introduces the model and concept of Futile Cycle. Moreover, the presentation of two completely different versions of the model, one which involves addition/removal of PTMs and the other formation/dissolution of GTF condensates we hope will also improve the clarity for why we chose this name.

We fear that “cooperative promoter activation” would be confused with the cooperative binding of transcription factors, which explains hypersensitivity to changes in transcription factor concentration but not hypersensitivity to structural change affecting E-P contact frequency. Based on these comments, and related feedback from colleagues, we have added an additional section to explain why cooperativity among enhancer-contacts is indeed an alternative mechanism to achieve sensitivity to structural change but not consistent with published data. In contrast the models we describe are consistent with published data. Please see Figure 2E-G and pages 10-11.

8. At the beginning of the Discussion section authors state they will propose future experiments in each section. However, in some of the sections it is not clear what specifically authors are proposing. These suggestions should be made clearer.

We have added several additional experimental suggestions to the discussion, and elaborated on several of our previous ones. We have also updated the formatting of this presentation with clear paragraph separation of the proposed experiments. We thank the reviewer for this suggestion which we hope will substantially improve the interest of the manuscript.

9. If I understand the model correctly, the non-linearity arises because of the increase rate of tag addition when tag is already present. The authors then speculate histone modifications can be one such tag. However, there is only so many sites of modification at a promoter. Can the authors analyze how the possible range of tag densities affects performance of the model? Is the range required biologically plausible?

Yes, the reviewer understands the model correctly in terms of the origin of nonlinear response. The number of promoter tags or number of condensate molecules needed to achieve hypersensitive responses as a function of other model parameters is now discussed in Figure 3 and Figure 3 supplement 1 and Figure 5 supplement 1. The plausibility of the assumptions of tag number is now discussed in the Methods in our new section on Model Assumptions (see page 30-31).

In what we now call the “PTM version” of the model, in order to keep the number of parameters to a minimum, we did not set an explicit saturation point for deposition of tags, and instead let the removal rate set the effective maximum. In the condensate model, depletion of the pool of proteins below the saturation threshold for condensation sets the maximum number of molecules at the promoter. For simplicity, we model this effect with a fixed constant we call “max cluster size”.

The numbers used in these simulations, 10-40, likely do approach saturation of available substrate sites (though perhaps multiple tags, such as H3K4me and histone acetylation together accumulate to produce the stable active state). There are multiple histone tails available in a single nucleosome, which contain multiple potential modification sites for some of the tags which are found at promoters. The abundant lysine residues in histone tails can each accumulate 3 methyl tags (though the position of the lysine in the tail can have distinct effects on transcription probability). Many of the histone tagging enzymes will also add their modifications (like acetylation) to other non-histone promoter-associated proteins. Alternatively, a greater number of enzyme states, for example, achieved by an enzyme whose stability of association with the promoter-complex increases steadily with additional PTMs, would allow a very hypersensitive response to emerge even with few total tags in the ON state.

10. Can the authors do more analysis to explore how rapid changes in gene expression may occur (e.g. upon signaling a gene may go up within minutes)? How much more frequent does the E-P interaction need to be for rapid switch to the active promoter state? Can the authors do an analysis where they change the rates of the futile cycle upon some signal: at what time scale does transcription then change (keeping E-P frequency the same)?

The temporal dynamics for a promoter that starts in an off-state, to switch to an on state, given a particular rate or E-P contact, is now shown in Figure 5e. The relation in the amount of time expected for transcription to change when E-P contact frequency is perturbed moderately, compared to the amount of time required for enhancer mediated gene activation is discussed on page 19. The relationship of these model properties to previous biochemical measurements is described in the discussion on page 26.

Briefly:

The absolute kinetic properties for the molecular species in our models are not known (how many acetyl or methyl groups per minute are added or removed from a promoter? How many GTFs condense or redissolve per minute?). We have also not taken the molecular specification of the model to such as an extreme as insisting that the ‘tag’ be histone-acetylation rather than histone methylation or PolII tail-phosphorylation, or insisting the condensate be made of PolII rather than Mediator—and the rate constants may be different for the different molecular species. Thus the model is of more use in understanding relative differences in response time between different mechanisms (e.g. enhancer activation vs. loop disruption), rather than providing precise predictions in absolute time.

However, we can and do compare relative temporal responses to different perturbations. In particular, for the discussion related to the timescales of cohesin depletion vs. TAD border deletion, we determine the amount of time it takes the promoter to switch off upon enhancer deactivation (in arbitrary units of model time) and compare this to the amount of time it would take the promoter to switch off if the E-P contact frequency was modulated by a factor of two (see Figure 5F,G). For a more comprehensive parameter sweep, we also plot the promoter state as a function of time for a whole parameter scan of E-P contact frequencies and different lag times (see Figure 5E).

From these analyses we concluded, if typical promoters take minutes to an hour to respond to enhancer activation/deactivation, then a few hours is insufficient time for those same promoters to respond to a small structural change (a 2 fold reduction in enhancer activity), and that much greater transcriptional changes would be observed at time scales orders of magnitude longer than the typical time for promoter activation/deactivation via enhancer activation/deactivation.

11. Due to the lack of description, in many sections it is not clear what are the specific inputs and outputs of the model (e.g. Figure 2).

This should now be clear. In short, the inputs are generally E-P contact frequency and the output is transcription rate, which we simplify by assuming it is equivalent to “condensate size” or “tag count”—as all the interesting behaviors of our model are already established by these parameters, there is was no need to add gratuitous complexity. See also responses to Major points 1 and 4 above.

12. The Methods section describes the chemical kinetics of the suggested reactions and the insulation score calculations. But it is not clear how do these inform each other, how are contact-frequency maps chosen/computed and cross-referenced with the local E-P kinetics?

The Methods section has been extended to describe each simulation presented.

Insulation score is a measure of relative contact frequency within a TAD vs. between TADs. Since much of the experimental data involves perturbations to TAD boundaries resulting in partial or complete merger of the TADs, the insulation score provides an upper bound on the magnitude of the change in contact frequency. It is because insulation scores are rarely larger than 2 that we repeatedly focus on 2 fold changes in contact frequency in our simulations. The calculation of insulation score is presented in the methods so the reader knows precisely where the claim that almost all TAD boundaries correspond to factor-of-2 or less differences in contact frequency.

13. In 587-588, the index of k is 2(n+1), which equals to 2n+2, but then in the next line the following assumption is made 2n+1 → n+1.

We apologize for the original version of this section, which included a confusing and unnecessary renumbering of the variables indexed by *k*. It has been completely rewritten and we hope the new version will be more readable.

14. The authors make assumptions that their kinetic considerations hold for n>2. What is the evidence?

For simplicity we performed the PTM model simulations with *n*=2. All of the ‘paradoxes’ discussed can be resolved with a PTM model in which only two stimulated states are needed.

An analysis of the continuum limit version of the PTM model shows that the effective Hill coefficient of the tag-addition process is set by the number of states. We presented the derivation for *n*=2, a straightforward reproduction of this derivation with *n*=3 or *n*=4 leads to the appearance of a tag-addition curve where the highest order polynomials are now *n*=3 and *n*=4 respectively. Choosing rate constants such that these terms dominate leads to a case where tag-addition converges to a Hill Function with Hill Coefficient *n*=3 or 4 (the less these terms dominate the flatter the effective Hill Coefficient). The effects of changing the steepness (i.e. effective Hill Coefficient) of the tag-addition curve is shown in Figure 3 supplement 1 and Figure 5 supplement 1. A detailed analysis of the effect for all options between *n*=1 and the expected behavior in the limit n-> infinity is now discussed in more detail in the section “Analytic inference of the parameter space of the PTM model” in the methods, see page 31-32.

Additionally, we presented a condensate model in which no enzymes are involved at all.

15. The language of the paper is often not technically precise with qualifiers missing, which could lead to ambiguities and misinterpretations. Here are some examples:

We have carefully revised the text for clarity of language.

p. 1, line 10, "difference in contact across TAD borders is usually less than twofold";

In this sentence, we highlight a qualitative difference in how structure changes (less than two fold), to how transcription changes (as much as ten fold, sometimes more). We find this qualitative sentence informative and readable, and supported by substantial evidence in the literature. Later in the text we provide explicit and precise quantitative justification for the statement. For example, Figure 1 supplement 4 plots the difference in contact across ALL TAD borders in the human genome from IMR90 cells using the Hi-C data from Rao et al. 2014, and shows that these insulations scores are rarely greater than 2 (less than 10% of TAD borders in that data have a score >2, the median score is ~1.32).

We feel it would unnecessarily interrupt the readability of the manuscript to say that “in IMR90 cells, 95% of TAD borders demarcate differences of less than 2 fold change in contact frequency”. It would also give the naive reader the impression this property is specific to IMR90 cells rather than generally true of available Hi-C data across the cell types and species which have been analyzed. We appreciate that this line in the abstract would feel better justified if it carried a citation, however *eLife* does not use citations in the abstract.

p. 1, line 17, "results from recent cohesion disruption";

We have added the citation to the ‘recent result’.

p. 2, line 71, "A simple model of hypersensitivity to changes in contact frequency".

We have reworded this line for clarity.

16. On p. 13, line 483, authors define Ostwald ripening as given by weak multivalent interactions; however, Ostwald ripening is a thermodynamic process. In addition, they propose that liquid condensates become larger due to Ostwald ripening, but there are also other processes that may occur, such as coalescence of condensates, which would also lead to larger condensates.

This part of the text has been removed.

17. An alternative explanation for TAD-specific enhancer action is that an E-P interaction within a TAD (between two convergent CTCF sites), one that is brought about by extruding cohesin, is not equivalent to an interaction that occurs between two loci on either side of a CTCF site and that can be a random collision that is not mediated by extruding cohesin. In other words, two interactions can be of the same frequency but can be of a very different molecular nature. I agree that this model would not explain the results of the experiment where cohesin is acutely removed.

We agree. Indeed a number of additional mechanisms could be proposed to reconcile different subsets of the paradoxes we discuss.

18. In the beginning of the introduction the authors introduce TADS. I recommend that the authors present this in a more nuanced way: compartment domains also appear as boxes along the diagonal, an issue that has led some in the chromosome folding field to be confused. This reviewer believes TADS are those domains that strictly depend on cohesin mediated loop extrusion, whereas compartment domains are not. If the authors agree, perhaps they can rewrite this section?

Because the term “TAD” is frequently used in discussing the paradoxes we seek to address with our model, we find it necessary to introduce it to the text.

We appreciate the term has become controversial in the field. Here, we use the term TAD as it was originally defined when it was originally coined in the literature (in work one of our reviewers co-authored, and before any cohesin depletion experiments were done): linear stretches of the genome in which intra-domain contact is greater than interdomain contact—features which appear as boxes or triangles in the popularly used heatmap representation of these data.

We disagree with (re)*defining*TADs to be features created by loop-extrusion for the following reasons: The term was coined before cohesin depletion experiments, and has been extensively applied to describe results in many cell types and many experiments in which cohesin has not yet been perturbed. If we redefine TADs as the product of loop extrusion, the TADs of Dixon 2012 in IMR90 cells, the TADs of Lupianez 2015 in mouse limbs, can be called at best “prospective TADs”, *since it has not been shown that in these cells* or limbs that they arise from loop extrusion and are not also compartment boundaries. Indeed, several of the examples of TAD boundaries highlighted in Dixon 2012 and Nora 2012 and Lupianez 2015 *are also* compartment boundaries, and are at genomic positions which retain a detectable border in HCT116 and or mESC cells following cohesin depletion.

While it is useful to talk about ‘structural features created by cohesin’ we propose it would be much less confusing to just call those ‘structural features created by cohesin (SFCCs ?), rather than co-opting the word TAD to this new purpose. Nearly a decade of discussing high resolution Hi-C maps, since Dixon 2012/Nora 2012, has demonstrated it is also useful to have a word to refer to boxes or triangles that we see on these maps, regardless of what we know of the molecular mechanisms that gave rise. The term TAD was introduced to do this in 2012 and seems to serve the purpose well, and we continue to use it as such.